# STEP-VQ: Frame-level Inference with VQ-VAE for Model-based Reinforcement Learning

## Abstract

Model-based reinforcement learning (MBRL) from pixels often encodes frames into discrete latent variables that form tokens for sequence model backbones to learn world model dynamics. Previous work adopts two main approaches, each facing distinct limitations. Categorical bottlenecks enable fast frame-level prediction by flattening spatial features into categorical distributions, but suffer from explosive parameter growing with resolution and code dimension. Conversely, vector-quantised variational autoencoder (VQ-VAE) methods achieve parameter efficiency through codebook quantisation but require slow token-level autoregressive prediction within frames, shifting computational complexity to the dynamics model and producing longer sequences that slow training and inference.

We propose STEP-VQ, a novel frame-level VQ-VAE-based world model that enables prediction of entire frames through single forward passes. STEP-VQ follows the latent-imagination paradigm with two components: a world model (VQ-VAE + sequence model) and a behaviour policy. The approach is sequence-model agnostic, working with both Mamba-2 and Transformer architectures without modifications. Our key insight is that fine-grained spatial structure preservation may be unnecessary for effective behaviour learning in latent space, as temporal dynamics can implicitly capture spatial patterns through frame-level prediction. We provide rigorous theoretical analysis grounded in variational inference, showing how our training objective emerges from evidence lower bound (ELBO) optimisation and why Kullback–Leibler (KL) divergence formulations enable superior performance through bidirectional optimisation.

On Atari-100k, STEP-VQ achieves competitive performance whilst dramatically improving efficiency: 11.2× faster training than a strong VQ-VAE based baseline, 4× parameter reduction compared to categorical bottlenecks, and growing advantages at higher resolutions (+27.4% mean improvement at 96×96). STEP-VQ reaches superhuman performance on 9 games versus 8 for categorical methods, with KL divergence providing 24.5% improvement over cross-entropy baselines. These results demonstrate that frame-level discrete quantisation offers a practical path to efficient, scalable MBRL using modern sequence architectures.

## 1 Introduction

Model-based reinforcement learning (MBRL) significantly improves sample efficiency, which has been a key challenge in applying reinforcement learning (Hafner et al., 2020; 2021; Kaiser et al., 2020; Hafner et al., 2023; Wang et al., 2024). One effective approach is to train a world model using real environment interactions (Ha & Schmidhuber, 2018; Hafner et al., 2019). Subsequently, instead of policy learning relying solely on costly real-world interactions, the world model can be used to generate 'imagined' rollouts used to refine both policy and value estimates. This allows each real interaction to yield many learning updates. The result is faster learning particularly in domains where data is scarce, expensive, or risky — such as robotics, healthcare, and industrial control — while still achieving strong final performance.

In MBRL, a central challenge is to learn compact, predictive state representations directly from high-dimensional pixel frames. Discrete latent spaces address this by compressing sequences of

observations into sequences of symbolic codes (Oord et al., 2017; Esser et al., 2021; Yu et al., 2022) that simplify dynamics learning and control, enabling human-level or superhuman performance on Atari, MuJoCo, and Crafter (Hafner et al., 2021; Wang et al., 2024; Dedieu et al., 2025). The key insight is that effective world models require alignment between visual encoding (how to discretise observations) and temporal prediction (how to model dynamics). Prior work learns such discrete latents via **Categorical bottlenecks (CB)** (e.g., the Dreamer family) (Hafner et al., 2021; 2023; Wang et al., 2025; Zhang et al., 2023) or **Codebook-based vector quantisation (VQ)** VQ-VAE–style codebooks (e.g., IRIS) (Oord et al., 2017; Micheli et al., 2023; 2024; Dedieu et al., 2025). However, these existing approaches each have their own caveats: CB suffers from parameter counts that grow with image resolution and code size while VQ suffers from slow, sub-frame token-level autoregressive rollouts. For example, IRIS connects a VQ codebook to a transformer-based world model but incurs prohibitively slow training due to per-token autoregressive prediction within frames (Micheli et al., 2023). Building on this, Parallel Observation Prediction (POP) and $\Delta$-IRIS improved training throughput yet still fall short of the efficiency of categorical bottlenecks (Cohen et al., 2024; Micheli et al., 2024). This computational bottleneck has limited the practical adoption of VQ-based world models despite their parameter efficiency advantages. To address this fundamental efficiency-performance trade-off, we target a hybrid approach using discrete, frame-based token representation (as in CB) with VQ-VAE's parameter efficiency, whose dynamics can be learnt and rolled out using frame-level predictions. The architecture of these two discretisation approaches is shown in Figure 1.

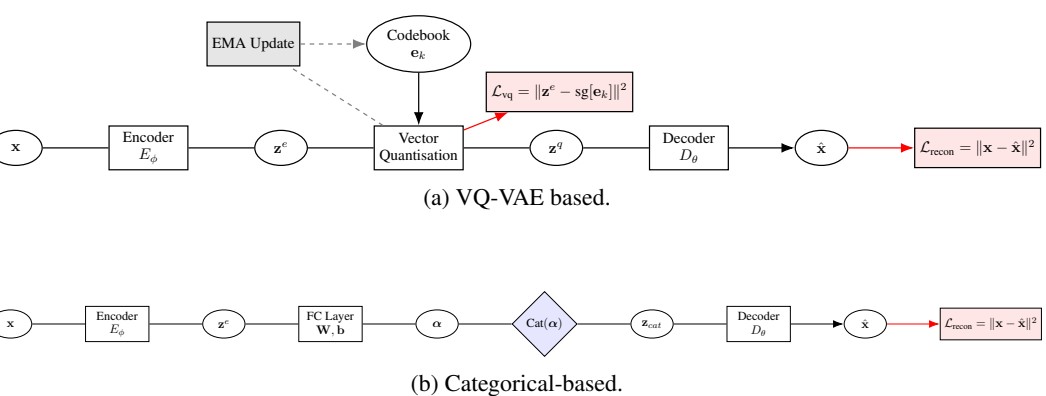

(a) VQ-VAE based.

(b) Categorical-based.

Figure 1: Two alternative tokeniser architectures: (a) VQ-VAE based; (b) Categorical bottlenecks based.

These design choices reflect different priorities in the efficiency-fidelity trade-off (as we show later in Figure 2 in Section 3). Categorical bottlenecks achieve fast frame-level prediction by mapping spatial features to high-dimensional categorical distributions that produce single tokens per frame, but this approach (i) discards spatial structure through flattening and (ii) creates explosive parameter scaling proportional to height$\times$ width$\times$ channels$\times$ code dimension$\times$ code class, inflating memory and latency as resolution or code cardinality grows. Conversely, VQ-VAE-based models preserve spatial structure through codebook quantisation and maintain parameter efficiency, but shift computational complexity to the dynamics model, requiring sequence modelling to process multiple tokens per frame rather than single tokens, producing longer sequences and slower training/inference.

However, our approach takes advantage of the observations that, in MBRL contexts, whilst autoregressive token-level prediction respects fine-grained spatial structure, this precision may not be necessary for effective behaviour learning. Adjacent frames often exhibit high redundancy, suggesting that building spatial understanding by exploiting temporal similarity patterns may be sufficient for behaviour models operating in latent space. This insight motivates our approach: frame-level prediction that leverages temporal dynamics to implicitly capture spatial structure whilst achieving computational efficiency. Therefore, we propose STEP-VQ (Single-pass spaTio-tEmporal Prediction with VQ-VAE), a training scheme that enables a sequence model to predict all discrete video codes in a single forward pass while enforcing spatiotemporal consistency. STEP-VQ follows the latent-imagination paradigm with two components: a world model (enhanced VQ-VAE autoencoder and sequence model) and a behaviour policy. The approach requires no modifications to

the sequence model: it can be dropped into any Dreamer-like architecture by simply replacing the categorical-projection layer, and it works with Mamba, or Transformers. Our main contributions are:

- **Frame-wise prediction for VQ-VAE world models.** We propose a training scheme that enables *frame-level* prediction of discrete codes in VQ-VAE–based world models, avoiding sub-frame token-level autoregression, therefore increasing the "imagination" speed.
- **Architecture-agnostic sequence modeling.** Empirically, the method is *sequence-backbone agnostic*: we show empirically that it works out of the box with both Transformers and Mamba-2, yielding comparable performance in the low-resolution domain without architecture-specific tuning.
- **Resolution-scalable performance at fixed model size.** STEP-VQ leverages VQ-VAE's codebook efficiency where parameters grow significantly slower than categorical bottle-necks as resolution increases, avoiding the explosive parameter scaling of CB methods that leads to performance degradation. At higher resolutions, STEP-VQ's advantages become more pronounced, reaching +27.4% mean improvement at 96×96 resolution.
- **Theoretical analysis of frame-level temporal prediction.** We provide rigorous theoretical foundations explaining how temporal redundancy enables frame-level VQ prediction to match autoregressive performance whilst achieving superior computational efficiency, grounded in variational inference principles.

## 2 STEP-VQ Architecture

STEP-VQ follows the **latent-imagination paradigm** for MBRL, similar to methods like Dreamer (Hafner et al., 2021), DRAMA (Wang et al., 2025), and STORM (Zhang et al., 2023). Our overall architecture comprises two main components:

1. **World Model**: An VQ-VAE based autoencoder that learns discrete latent representations, paired with a sequence model (Mamba-2 or Transformer) that captures temporal dynamics.
2. **Behaviour Model**: A standard actor-critic model $\pi(a_t \mid h_t, z_t)$ that operates directly on the state predicted by $h_t$ and the current latent codes $z_t$.

The key innovation lies in enabling frame-level prediction for VQ-VAE methods—previously forced into slow sub-frame token-level autoregression while retaining the training efficiency and scalability advantages of CB-based methods. This approach enables the behaviour model to use latent variables directly as they represent entire frames rather than individual spatial tokens by exploiting temporal redundancy in images. We provide theoretical foundations leveraging variational inference principles, temporal redundancy exploitation analysis, and novel training procedures to ground our loss function choices and explain when frame-level prediction can match autoregressive performance.

Crucially, since the behaviour policy operates only on latent codes rather than reconstructed images, reconstruction quality drives representation learning rather than direct policy optimisation as shown in Figure 6.

### 2.1 World Model Implementation

The following subsections detail the technical implementation and theoretical framework that enables VQ-based frame-level prediction. We consider a partially observable Markov decision process (POMDP). At each discrete time $t$, the agent receives an image observation $x_t \in \mathbb{R}^{3 \times H \times W}$ and selects an action $a_t \in \mathcal{A} = \{0, 1, \ldots, n\}$ where $n$ is the number of available (task-dependent) actions.

Given a trajectory $(x_{1:T}, a_{1:T-1})$, a 2D encoder $\text{Enc}_\phi$ produces continuous latent features: $z_t^e = \text{Enc}_\phi(x_t) \in \mathbb{R}^{H' \times W' \times D}$ where $H', W'$ are the spatial dimensions and $D$ is the embedding dimension of the latent space. A vector-quantisation codebook $E = \{e_1, \ldots, e_C\} \subset \mathbb{R}^D$ (matrix $E \in \mathbb{R}^{C \times D}$) defines a per-patch quantiser over $(i, j) \in \{1, \ldots, H'\} \times \{1, \ldots, W'\}$: $k_{t,ij} = \arg\min_{c \in [C]} \|z_t^e[i, j] - e_c\|_2^2$, yielding the discrete index map $z_t = \{k_{t,ij}\}_{i,j} \in [C]^{H' \times W'}$ and its embedding $z_t^q = E[z_t] \in \mathbb{R}^{H' \times W' \times D}$ with $z_t^q[i, j] = e_{k_{t,ij}}$.

Because quantisation is applied per spatial site $(i, j)$, this discretisation preserves the $H' \times W'$ lattice, maintaining spatial locality and structure (without flattening). Training follows VQ-VAE practice: gradients are passed through the quantiser via a straight-through estimator with a commitment loss, and the codebook is updated with the exponential moving average (EMA) of the encoder embeddings; the decoder consumes the gathered embeddings $z_t^q$ in place of the continuous latents $z_t^e$. Concretely, we use the standard VQ-VAE commitment loss

$$\mathcal{L}_{\text{vq}} = \beta \left\| z_t^e - \text{sg}[z_t^q] \right\|_2^2, \quad \beta > 0, \tag{1}$$

where $\text{sg}[\cdot]$ denotes the stop gradient operation. The 2D decoder likelihood remains $p_\theta(x_t \mid z_t) \equiv p_\theta(x_t \mid z_t^q)$.

For context, we briefly compare our VQ-VAE approach with CB alternatives. Categorical methods represent each frame by a fixed number $M$ of categorical positions, each taking one of $K$ symbols. The flattened encoder latents are mapped to $\alpha_t \in \mathbb{R}^{M \times K}$ logits, with $M$ categorical distributions yielding indices $k_t \in [K]^M$. This approach discards spatial locality and introduces parameters proportional to $H'W'D \cdot (M \cdot K)$, making it sensitive to image resolution. Variants are used in STORM (Zhang et al., 2023), Dreamer (Hafner et al., 2021; 2023), and DRAMA (Wang et al., 2025), and serve as our experimental baselines in Section 3.2.

### 2.1.1 Sequence Model Component

The sequence model captures the temporal dynamics of the environment. Crucially STEP-VQ is sequence-model agnostic, meaning that it does not mandate the use of a specific sequence-model architecture. The sequence model learns to predict future discrete codes $z_{t+1}$ given the historical context and actions. We demonstrate STEP-VQ's versatility by evaluating its performance with two popular modern sequence modelling architectures: Mamba-2 and Transformer, with detailed comparisons presented in Section 3.

## 2.2 Training Procedure

Training uses batches of real frames from replay data in shape $(B, L, H, W, 3)$, where $B$ is the batch size and $L$ is the sequence length. The encoder and VQ layer produce quantised codes $z_t^\star$ (ground truth) representing the temporal sequence in shape $(B, L, H', W')$, where $H', W'$ are the spatial dimensions after encoder downsampling. These codes, along with actions, are processed by the sequence model to generate recurrent states $(B, L, d_h)$, where $d_h$ is the recurrent hidden dimension, which predict future codes $\hat{z}_t$ (predictions). Training aligns the predicted codes (student) with encoder-assigned codes (teacher), where $z^\star[:, 1 : L]$ serves as the teacher and $\hat{z}[:, : L-1]$ serves as the student, implementing temporal offset for next-step prediction.

## 2.3 Mathematical Framework

Having introduced the VQ-VAE autencoder and sequence model components, we now address the key challenge: how to train the sequence model to predict discrete codes for frame-level prediction effectively. We frame this as a problem of alignment between visual encoding and temporal prediction.

### 2.3.1 Spatial Dynamics and Temporal Redundancy

We now provide the mathematical formulation underlying the training procedure. The encoder $q_\phi(z_t \mid x_t)$ acts as a **posterior distribution** that assigns discrete codes given visual observations:

$$q_\phi\big(z_t[i, j] = c \mid x_t\big) = \frac{\exp\big(-\beta \left\| z_t^e[i, j] - e_c \right\|^2\big)}{\sum_{k=1}^{C} \exp(-\beta \left\| z_t^e[i, j] - e_k \right\|^2)}. \tag{2}$$

where $\beta$ controls the temperature (standard VQ-VAE uses $\beta \to \infty$ for hard assignments).

The temporal dynamics are modelled through a recurrent state $h_t \in \mathbb{R}^{d_h}$ that summarises the history of latent codes and actions up to time $t - 1$:

$$h_1 \sim p_\psi(h_1), \qquad h_t = f_\psi(h_{t-1}, z_{t-1}, a_{t-1}) \quad \text{for } t \geq 2, \tag{3}$$

so that $h_t$ is a deterministic function of $(z_{1:t-1}, a_{1:t-1})$. The dynamics model $p_\psi(z_t \mid h_t)$ serves as a **prior distribution** that predicts codes given this encoded history, with the equivalence $p_\psi(z_t \mid h_t) \equiv p_\psi(z_t \mid z_{1:t-1}, a_{1:t-1})$. This prior captures temporal dependencies whilst the posterior captures visual content.

Our dynamics model $p_\psi(z_t \mid h_t)$ approximates the standard spatial categorical distribution:

$$p_\psi(z_t \mid h_t) \approx \prod_{i=1}^{H'} \prod_{j=1}^{W'} \mathrm{Cat}\big(z_t[i,j]; \mathrm{softmax}\big(f_\psi(h_t)[i,j]\big)\big). \tag{4}$$

This approximation exploits **temporal redundancy**—enabling frame-level prediction when temporal patterns contain sufficient mutual information to predict spatial dependencies. The approach works when temporal correlations ($z_t^{(i,j)}$ from $z_{t-1}^{(i,j)}$) are more predictive than spatial correlations, with achievable approximation quality given using mutual information bounds in Section B.2.

### 2.3.2 TRAINING LOSS FORMULATION

The complete training objective combines reconstruction, VQ-VAE commitment, and dynamics losses:

$$\mathcal{L}_{\text{total}} = \mathcal{L}_{\text{recon}} + \mathcal{L}_{\text{vq}} + \mathcal{L}_{\text{dynamics}}. \tag{5}$$

The dynamics loss aligns predicted categorical distributions with ground-truth code assignments, as derived from ELBO principles in Section B.1. During training, the sequence model predicts $\hat{z}_{t+1}$ from temporal history $h_t$, supervised against ground-truth codes $z_{t+1}^\star$ from the encoder. The loss formulation becomes:

$$\mathcal{L}_{\text{dynamics}} = -\sum_{i,j} \log p_\psi\big(\hat{z}_{t+1}[i,j] = z_{t+1}^\star[i,j] \mid h_t\big). \tag{6}$$

where the sequence-model follows the encoder in the distributional alignment described above, with averaging over batch and sequence dimensions in practice. This is precisely the **cross-entropy loss** between the predicted categorical distributions and ground-truth code assignments.

**KL Divergence and Practical Implementation.** Empirically, KL divergence loss outperforms cross-entropy through superior training methodology. Rather than hard assignments from standard VQ-VAE, we use soft posterior targets based on encoder distances to all codebook entries. The KL framework enables bidirectional training with stop gradients that jointly optimise both the encoder and dynamics model, preventing model collapse whilst enabling advanced techniques like free-bit (Kingma et al., 2016) for maintaining representation quality. The practical loss formulation is:

$$\mathcal{L}_{\text{KL}} = \lambda_1 D_{\text{KL}}\big[\mathrm{sg}(p_\psi(\hat{z}_{t+1} \mid h_t)) \,\|\, q_\phi(z_{t+1}^\star \mid x_{t+1}^\star)\big] + \lambda_2 D_{\text{KL}}\big[p_\psi(\hat{z}_{t+1} \mid h_t) \,\|\, \mathrm{sg}\big(q_\phi(z_{t+1}^\star \mid x_{t+1}^\star)\big)\big].$$

This not-only prevents model collapse but enables complementary teacher-student learning between encoder and dynamics model, with detailed derivation provided in Section B.3.

## 3 EVALUATION

We evaluated STEP-VQ on the Atari 100k benchmark, which limits agents to 100k environment interactions representing approximately 2 hours of real-time play (Kaiser et al., 2020; Machado et al., 2018). This benchmark is widely used for assessing MBRL methods as it emphasizes sample efficiency. We follow the standard evaluation protocol and report per-game human-normalised scores (HNS):

$$\mathrm{HNS} = \frac{\mathrm{score}_{\text{agent}} - \mathrm{score}_{\text{random}}}{\mathrm{score}_{\text{human}} - \mathrm{score}_{\text{random}}}. \tag{7}$$

over the standard 26-game set (Kaiser et al., 2020; Schwarzer et al., 2023). Our codebase builds on the DRAMA MBRL implementation (Wang et al., 2025). Unless otherwise stated, results use the configuration shown in Section F for the Mamba2-based architecture (STEP-VQ-M) and the Transformer-based architecture (STEP-VQ-T).

The primary innovation of STEP-VQ addresses the computational bottleneck that has limited VQ-based world models. We therefore begin by evaluating the training efficiency gains that enable scalable VQ-based MBRL.

## 3.1 TRAINING EFFICIENCY: ADDRESSING VQ-BASED TRAINING BOTTLENECKS

A significant advantage of STEP-VQ is its training efficiency compared to existing VQ-based methods. Our timing comparison, on a single L40S GPU, demonstrates substantial speedups across configurations (Figure 2). STEP-VQ-IRIS, a version of STEP-VQ using IRIS-consistent hyperparameters and a transformer sequence model for fair comparison, trains 11.2× faster than IRIS. More importantly, STEP-VQ-T, the configuration used for performance evaluation in Table 3, achieves 9.7× speedup while maintaining equivalent performance.

The efficiency bottleneck in IRIS stems from token-level autoregression: 20-frame sequences result in 20×16=320 actual tokens to be processed due to spatial tokenisation, with batch size 64 and smaller models (256 dimensions, 10 layers). STEP-VQ eliminates this bottleneck through frame-level prediction enabled by spatial independence, allowing longer temporal context (128 frames in STEP-VQ-T) and different architectures (512 dimensions, 4 layers) without proportional computational penalty. The fact that STEP-VQ-T uses 6.4× longer sequences whilst maintaining 9.7× speedup demonstrates the scalability advantages of frame-level approaches.

By enabling frame-level prediction in a single forward pass, STEP-VQ eliminates the token-level autoregressive bottleneck whilst retaining VQ-VAE's parameter efficiency and preserving spatial structure. This advancement makes VQ-based MBRL competitive for applications requiring efficient temporal modelling with extended temporal context.

Having demonstrated substantial improvements in training efficiency, we next evaluate whether these gains come at the cost of performance quality.

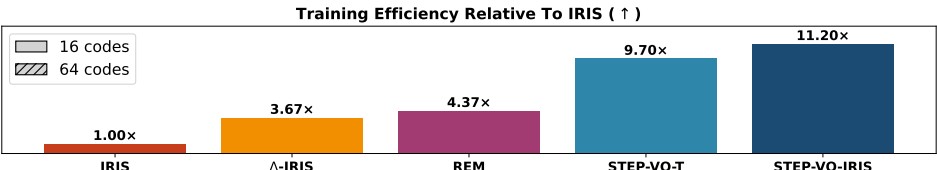

(a) Training efficiency comparison between STEP-VQ and other VQ-VAE-based models.

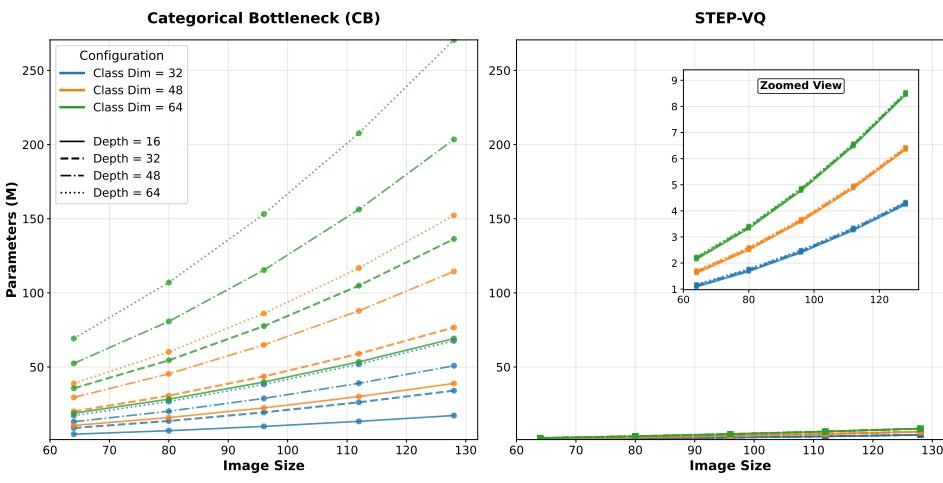

(b) Parameter efficiency analysis between CB and STEP-VQ.

Figure 2: Model efficiency and parameter analysis for STEP-VQ compared to baseline methods.

## 3.2 Overall Performance: Comparison with Categorical Bottlenecks (CB)

As established in Section 2.3, approximating spatial structure through temporal dynamics is theoretically sound for MBRL applications. As CB methods use independent sampling that completely discards spatial structure in exchange for computational efficiency, we empirically evaluate against CB-based approaches to validate this theoretical framework. To do so, we replace the VQ-VAE based autoencoder in our Mamba- and Transformer-based implementation of STEP-VQ (referred to as CB-M and CB-T - with detailed configurations provided in Section F). This comparison tests whether our spatial independence approximation, which preserves more spatial information than CB's complete spatial flattening, can achieve competitive performance whilst enjoying significantly higher training efficiency than VQ-based world models such as IRIS. We use CB-based architectures as our primary baseline whilst using original IRIS results for Atari 100K as an additional reference point. We note that re-evaluating IRIS on the Atari 100k benchmark would have required prohibitive computational resources.

All method variants (STEP-VQ-M, STEP-VQ-T, CB-M, CB-T) use identical hyperparameters across all Atari games. STEP-VQ demonstrates overall higher mean performance for both architectures tested, evaluated using 3 seeds with performance averaged across 10 episodes per game. Significantly, STEP-VQ consistently achieves superhuman performance (HNS > 1) on 9 games versus 8 for the CB baselines across both architectures. With Mamba2, STEP-VQ-M achieves a mean HNS of 1.110 compared to 1.053 for categorical quantisation (+5.4% improvement, Table 1). With Transformer, STEP-VQ-T scores 1.048 versus 1.012 for categorical bottlenecks (+3.5% improvement, Table 2). The overall training curve is shown in Figure 4.

For additional context with VQ-based methods, STEP-VQ-T achieves equivalent mean HNS (1.048) and higher median scores compared to published IRIS results (Table 3, with IRIS results marked * indicating external results from the original paper). This competitive performance is notable given that STEP-VQ-T uses fixed hyperparameters for all games compared to IRIS, which has game-specific hyperparameter tuning for `Freeway`. However, readers should interpret this comparison carefully as IRIS results are from their original publication rather than reproduced in our environment.

These results must be interpreted within the context of parameter efficiency: STEP-VQ achieves these competitive results using significantly fewer parameters (2,175,616) compared to the CB baselines (8,919,552), representing a 4× parameter reduction. Whilst this efficiency trade-off contributes to mixed individual game performance and lower median scores in some comparisons, it becomes increasingly valuable at higher resolutions (detailed in Section 3.3) where categorical methods experience parameter explosion as shown in the Figure 2. These results highlight STEP-VQ's architecture-agnostic benefits and hyperparameter robustness whilst showing varying performance across individual games, which we interpret to be due to varying degrees of temporal redundancy exploitation across game types, with performance correlating to how well temporal patterns can substitute for explicit spatial modelling as analysed in Section B.2.

Building on both efficiency and performance advantages, we evaluate how these benefits translate to scalability at higher resolutions.

## 3.3 Resolution Scalability: Enhanced Performance at Higher Resolutions

STEP-VQ's advantages become more pronounced at higher resolutions, demonstrating superior scalability compared to CB architectures. At 96×96 resolution, STEP-VQ's performance advantage over categorical quantisation grows to +27.4% mean improvement (+35.9% median) as shown in Table 5 and Figure 3.

To ensure unbiased evaluation, we tested 8 games selected using three criteria based on performance patterns in Table 1: (1) high-influence games that significantly impact overall metrics (`Krull`, `Boxing`, `Jamesbond`, `RoadRunner`), (2) games where categorical methods achieve superhuman performance whilst STEP-VQ does not (`UpNDown`, `BankHeist`), and (3) games where STEP-VQ achieves superhuman performance whilst categorical methods do not (`Breakout`, `Assault`).

The scalability advantage is evident in performance degradation analysis: whilst both methods experience performance reduction at higher resolution, STEP-VQ decreases by 7.5% (Table 6) compared to categorical methods' 19.6% degradation (Table 7). At 96×96 resolution, STEP-VQ wins 6

out of 8 games and maintains superior mean and median performance across the diverse scenarios tested. This suggests STEP-VQ's codebook-based quantisation scales more effectively than categorical bottlenecks for higher-resolution processing. An example of the quantised codes of STEP-VQ at 96×96 resolution is shown in Figure 9.

Finally, we validate our theoretical analysis by directly comparing the two loss functions derived from our ELBO framework.

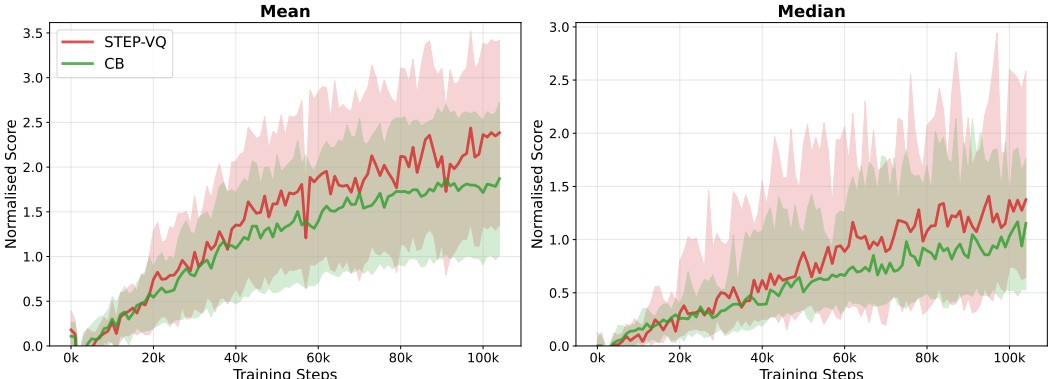

Figure 3: Higher resolution (96×96) performance comparison across 8 systematically selected games. STEP-VQ demonstrates superior scalability with +27.4% mean advantage (+35.9% median) over categorical quantisation, whilst showing reduced performance degradation (7.5% vs 19.6% for categorical methods). Games were selected across three performance regimes to ensure unbiased evaluation: high-influence games (`Krull`, `Boxing`, `Jamesbond`, `RoadRunner`), categorical-advantage games (`UpNDown`, `BankHeist`), and STEP-VQ-advantage games (`Breakout`, `Assault`), demonstrating consistent scalability benefits across diverse scenarios.

### 3.4 CROSS-ENTROPY VS. KL DIVERGENCE: FROM THEORY TO PRACTICE

As established in Section 2.3.2, cross-entropy loss emerges naturally from the ELBO's prior-posterior alignment objective. Since both $p_\psi(z_t \mid h_t)$ and $q_\phi(z_t \mid x_t)$ are categorical distributions over discrete codes, their alignment during the imagination phase becomes:

$$\mathcal{L}_{\text{CE}} = -\sum_{t=1}^{L-1} \sum_{i,j} \log p_\psi(z_t[i,j] = q_\phi(z_t[i,j] \mid x_t^\star) \mid h_t). \tag{8}$$

This establishes cross-entropy as a natural objective for discrete codes. However, our empirical evaluation shows that KL divergence loss provides a **24.5% performance improvement** over the cross-entropy baseline.

We directly compared STEP-VQ using cross-entropy loss versus our proposed KL divergence formulation on the same 8 games used for resolution scalability analysis, with the same selection rationale to ensure unbiased evaluation. The results demonstrate general advantages for KL divergence across multiple metrics: mean HNS improves from 2.070 to 2.577, whilst median performance increases from 1.209 to 1.300 (Table 4).

The advantage varies across games: KL divergence shows substantial improvements in games like Krull (+45.0%) and Boxing (+27.1%), whilst cross-entropy performs competitively in games like UpNDown and Assault. This empirical validation supports our theoretical analysis that KL divergence enables more effective training.

## 4 RELATED WORK

Model-based reinforcement learning has emerged as a key approach to improve sample efficiency by learning environment dynamics for policy optimisation (Sutton & Barto, 1998; Hafner et al., 2020;

2021; Kaiser et al., 2020; Wang et al., 2024). Recent advances using discrete latent representations have achieved human-level performance across diverse domains (Hafner et al., 2021; Dedieu et al., 2025), with the central challenge being learning compact, predictive representations that enable efficient dynamics modelling whilst preserving visual information necessary for control. A critical design choice in this area concerns discrete representation learning approaches.

### 4.1 DISCRETE REPRESENTATION LEARNING

MBRL methods typically employ one of two discrete representation approaches: vector quantisation (VQ-VAE) and categorical bottlenecks, each facing distinct trade-offs.

VQ-VAE (Oord et al., 2017) provides parameter-efficient codebook representations and has been applied to world modelling in IRIS (Micheli et al., 2023), but requires slow token-level autoregressive prediction within frames. Improvements like $\Delta$-IRIS (Micheli et al., 2024) and POP (Cohen et al., 2024) enhanced throughput but remain computationally slower than categorical methods.

Categorical bottlenecks, popularised by Dreamer (Hafner et al., 2021; 2023) and variants like DRAMA (Wang et al., 2025) and STORM (Zhang et al., 2023), enable fast frame-level prediction but suffer from explosive parameter scaling proportional to image resolution and code dimensions.

The choice of sequence modelling architecture further influences these trade-offs.

### 4.2 ARCHITECTURAL FLEXIBILITY IN MBRL

Although $\Delta$-IRIS (Micheli et al., 2024) and POP (Cohen et al., 2024) have worked to make VQ-VAE-based world models faster, they still fall short of categorical bottleneck efficiency. Another limitation is that these approaches often require modifications to the sequence model architecture to achieve their improvements. Given that modern MBRL methods employ increasingly diverse sequence architectures—STORM uses Transformers (Vaswani et al., 2017) whilst DRAMA employs Mamba-2 (Gu & Dao, 2024)—such architecture-specific modifications limit flexibility and broader applicability.

STEP-VQ addresses these limitations by being sequence-model agnostic, working effectively with both Transformers and Mamba-2 without requiring architecture-specific modifications. Our approach enables frame-level prediction with VQ-VAE representations whilst maintaining the architectural flexibility needed for diverse MBRL applications.

## 5 CONCLUSION

This work addresses a fundamental limitation in MBRL: the computational bottleneck that has hindered VQ-VAE-based world models. STEP-VQ bridges VQ-VAE's parameter efficiency with categorical bottlenecks' computational speed, achieving $11.2\times$ training speedup over IRIS whilst maintaining competitive performance with $4\times$ fewer parameters. Our evaluation demonstrates advantages across different sequence-modelling architectures and growing benefits at higher resolutions (+27.4% at 96×96), with KL divergence providing 24.5% improvement over cross-entropy baselines.

Beyond empirical advances, this work provides a theoretical foundation for frame-level discrete prediction through variational inference principles. Our ELBO derivation establishes the mathematical basis for discrete dynamics learning, whilst spatial independence analysis explains when frame-level prediction can match autoregressive performance. The insight that temporal dynamics can implicitly capture spatial structure challenges conventional MBRL approaches and opens new research directions in efficient world model design. The architecture-agnostic framework enables broad applicability across modern sequence architectures.

While our evaluation focuses on Atari domains, the theoretical framework provides tools for broader application. The mutual information analysis offers principled approaches for adaptive spatial modelling, and the efficiency gains suggest potential for high-resolution, real-world applications. STEP-VQ represents a significant advancement toward practical, scalable MBRL that combines theoretical rigor with computational efficiency, offering a viable path for efficient world models using modern sequence architectures.

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

APPENDIX

# A  THE USE OF LARGE LANGUAGE MODELS (LLMs)

We use LLMs as a general-purpose assist tool for grammar and typo examination.

# B  THEORETICAL FOUNDATIONS

This appendix provides detailed mathematical derivations underpinning STEP-VQ's theoretical framework.

## B.1  ELBO DERIVATION FOR SEQUENTIAL DISCRETE LATENTS

We derive how cross-entropy loss emerges naturally from the ELBO in sequential discrete latent variable models.

**Variational Framework.**  Given trajectory $(x_{1:T}, a_{1:T-1})$, we seek to maximise:

$$\log p(x_{1:T} \mid a_{1:T-1}) \geq \mathbb{E}_{q_\phi(z_{1:T}|x_{1:T})} \left[ \log \frac{p_\theta(x_{1:T} \mid z_{1:T}) \cdot p_\psi(z_{1:T} \mid a_{1:T-1})}{q_\phi(z_{1:T} \mid x_{1:T})} \right]. \quad (9)$$

The dynamics term decomposes as:

$$p_\psi(z_{1:T} \mid a_{1:T-1}) = p_\psi(z_1) \prod_{t=2}^{T} p_\psi(z_t \mid h_t). \quad (10)$$

where $h_t$ encapsulates temporal history $(z_{1:t-1}, a_{1:t-1})$.

**Cross-Entropy Emergence.**  The dynamics loss becomes:

$$\mathcal{L}_{\text{dynamics}} = - \sum_{t=\tau+1}^{T} \sum_{i,j} \log p_\psi(z_t[i,j] = q_\phi(z_t[i,j] \mid x_t^\star) \mid h_t). \quad (11)$$

Since $p_\psi(z_t[i,j] \mid h_t) = \text{softmax}(f_\psi(h_t)[i,j,:])$ and $q_\phi(z_t[i,j] \mid x_t^\star)$ provides discrete targets, this is precisely cross-entropy loss between predicted distributions and ground-truth codes.

## B.2  SPATIAL INDEPENDENCE ANALYSIS

The factorisation $p_\psi(z_t \mid h_t) = \prod_{i,j} p(z_t[i,j] \mid h_t)$ assumes spatial conditional independence. This approximation quality can be bounded using mutual information.

**Approximation Error Bound.**  The error is small when:

$$I(z_t[i,j]; \ z_t[i',j'] \mid h_t) \approx 0 \quad \text{for } (i,j) \neq (i',j'). \quad (12)$$

This holds when temporal history $h_t$ contains sufficient information about spatial structure, making spatial locations conditionally independent given temporal context.

**Game-Specific Implications.**

- **Low MI games** (e.g., Breakout): Static backgrounds + deterministic physics $\rightarrow$ temporal history determines spatial layout $\rightarrow$ good approximation
- **High MI games** (e.g., exploration): Novel spatial structures unpredictable from actions $\rightarrow$ poor approximation

### B.3 KL DIVERGENCE VS CROSS-ENTROPY

The superior performance of KL divergence over cross-entropy stems from proper posterior formulation.

**Soft Posterior.** Instead of deterministic assignment $\delta(z_t - z_t^\star)$, the true posterior is:

$$q_\phi(z_t[i,j] = c \mid x_t) = \frac{\exp\big(-\beta \left\| z_t^e[i,j] - e_c \right\|^2\big)}{\sum_{k=1}^{C} \exp(-\beta \left\| z_t^e[i,j] - e_k \right\|^2)}. \tag{13}$$

**Bidirectional Training.** Practical implementation uses stop gradients:

$$\mathcal{L}_{\mathrm{KL}} = \lambda_1 D_{\mathrm{KL}}[\mathrm{sg}(p_\psi(z_t \mid h_t)) \parallel q_\phi(z_t \mid x_t^\star)] + \lambda_2 D_{\mathrm{KL}}[p_\psi(z_t \mid h_t) \parallel \mathrm{sg}(q_\phi(z_t \mid x_t^\star))]. \tag{14}$$

This prevents model collapse whilst enabling complementary learning between encoder and dynamics model.

## C  TRAINING CURVE

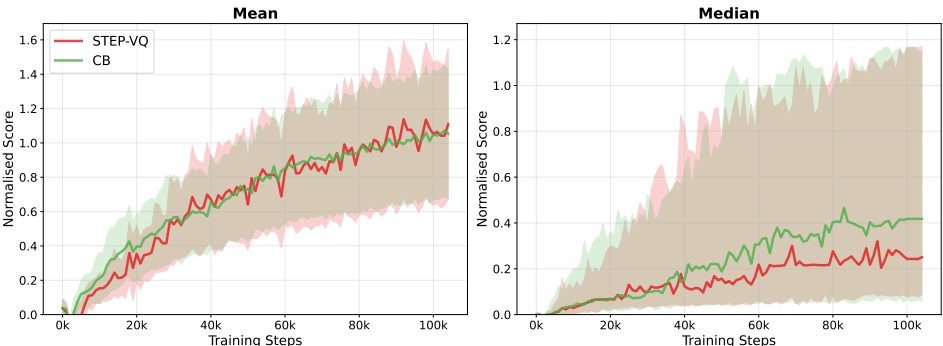

(a) Mamba2: STEP-VQ vs Categorical training curves over 100k environment steps, evaluated every 1k steps with 10-episode averaging at 64×64 resolution

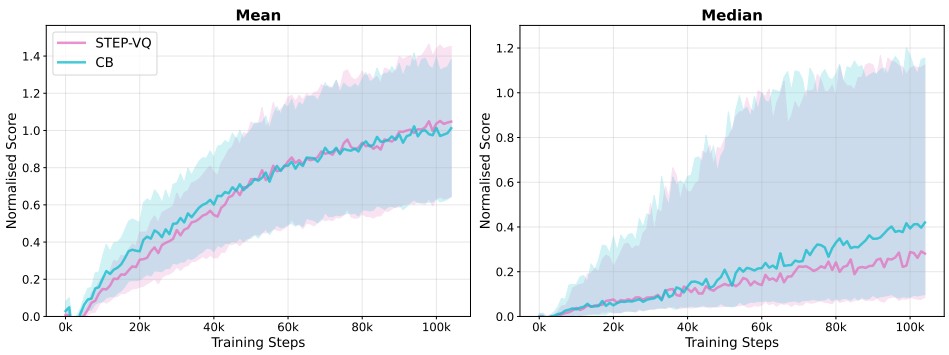

(b) Transformer: STEP-VQ vs Categorical bottlenecks training curves over 100k environment steps, evaluated every 1k steps with 10-episode averaging at 64×64 resolution

Figure 4: STEP-VQ vs. Categorical quantization performance across architectures on Atari 100k benchmark. Both Mamba2 and Transformer architectures show general improvements with STEP-VQ, with Mamba2 achieving higher absolute performance whilst results vary across individual games.

## D  STEP-VQ AND STANDARD VQ-BASED MBRL "IMAGINATION" PROCESSES DIAGRAM

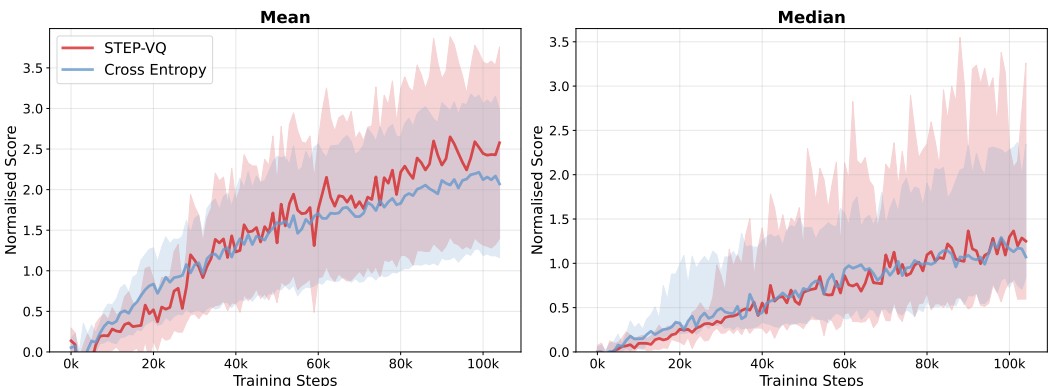

Figure 5: Loss function comparison: KL divergence vs cross-entropy for STEP-VQ across 8 systematically selected games, showing +24.5% mean improvement and +7.4% median improvement. Results demonstrate general advantages for KL divergence whilst acknowledging game-specific variation.

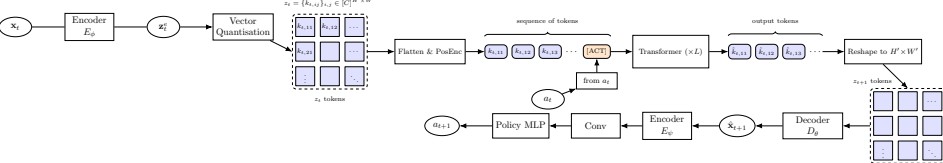

(a) Standard VQ-based MBRL "imagination" process.

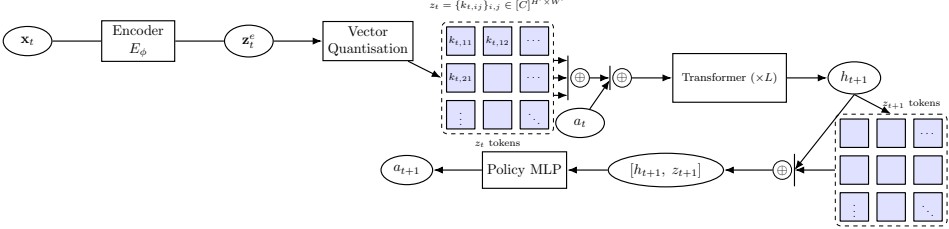

(b) STEP-VQ based MBRL "imagination" process.

Figure 6: STEP-VQ and standard VQ-based MBRL "imagination" processes. STEP-VQ enables frame-level prediction for VQ-VAE methods and the behaviour model operates directly on the latent codes rather than reconstructed images.

# E DETAILED PERFORMANCE RESULTS

This appendix provides comprehensive per-game performance comparisons across all tested architectures and quantisation methods.

| Game | STEP-VQ Mamba2 | CB Mamba2 | Difference | Change (%) |
|---|---|---|---|---|
| Alien | **0.142** | 0.027 | 0.115 | **+425.9** |
| Amidar | 0.073 | **0.082** | -0.009 | -11.0 |
| Assault | **1.054** | 0.737 | 0.317 | **+43.0** |
| Asterix | **0.154** | 0.140 | 0.014 | **+10.0** |
| BankHeist | 0.294 | **1.371** | -1.077 | -78.6 |
| BattleZone | 0.085 | **0.142** | -0.057 | -40.1 |
| Boxing | **7.244** | 6.169 | 1.075 | **+17.4** |
| Breakout | **1.071** | 0.199 | 0.872 | **+438.2** |
| ChopperCommand | **0.145** | 0.061 | 0.084 | **+137.7** |
| CrazyClimber | 2.622 | **2.998** | -0.376 | -12.5 |
| DemonAttack | 0.012 | **0.017** | -0.005 | -29.4 |
| Freeway | **0.911** | 0.378 | 0.533 | **+141.0** |
| Frostbite | 0.046 | **0.048** | -0.002 | -4.2 |
| Gopher | 0.564 | **0.745** | -0.181 | -24.3 |
| Hero | 0.217 | **0.415** | -0.198 | -47.7 |
| Jamesbond | **1.597** | 1.190 | 0.407 | **+34.2** |
| Kangaroo | 0.631 | **0.700** | -0.069 | -9.9 |
| Krull | **7.628** | 6.045 | 1.583 | **+26.2** |
| KungFuMaster | 1.144 | **1.253** | -0.109 | -8.7 |
| MsPacman | 0.207 | **0.304** | -0.097 | -31.9 |
| Pong | **1.167** | 1.162 | 0.005 | **+0.4** |
| PrivateEye | -0.000 | **0.025** | -0.025 | -100.0 |
| Qbert | 0.099 | **0.252** | -0.153 | -60.7 |
| RoadRunner | 1.528 | **1.945** | -0.417 | -21.4 |
| Seaquest | **0.012** | 0.008 | 0.004 | **+50.0** |
| UpNDown | 0.203 | **0.963** | -0.760 | -78.9 |
| **Overall Mean** | **1.110** | 1.053 | **0.057** | **+5.4** |
| **Overall Median** | 0.256 | **0.396** | -0.141 | -35.6 |

Table 1: Performance comparison between STEP-VQ and CB with Mamba2 on Atari 100K benchmark. Values show mean normalised scores across 3 seeds and evaluate the performance by averaging across 10 episodes. Best performing method per game is highlighted in bold. STEP-VQ has 9 superhuman games (HNS > 1) while CB has 8. Both STEP-VQ and CB have a close superhuman game: STEP-VQ's `Freeway` is 0.911 with the same hyperparameter as the other games while CB's `UpNDown` achieves 0.963.

## E.1 EXPLORATORY GAME PATTERN ANALYSIS

Preliminary analysis of game-specific performance reveals potential patterns that may inform future research directions, though this analysis is exploratory and post-hoc in nature. STEP-VQ shows stronger relative performance in six games: Alien (+279.4% average across architectures), Breakout (+265.3%), Freeway (+187.8%), Assault (+42.5%), Jamesbond (+25.8%), and Boxing (+16.5%).

These games appear to share certain characteristics such as continuous action requirements and spatial-temporal coordination challenges. For example, Breakout requires paddle positioning coordinated with ball trajectory, whilst Alien involves movement-shooting coordination. However, we

acknowledge this analysis is exploratory and requires careful interpretation given the small sample size and post-hoc nature.

Conversely, categorical quantisation shows advantages in other game types (BankHeist, Gopher, Up-NDown), potentially reflecting different computational requirements. Future work could investigate whether these patterns reflect systematic differences in how the two quantisation approaches handle different types of temporal dependencies, though such analysis would require larger game samples and controlled studies to establish causal relationships.

| Game | STEP-VQ Transformer | Categorical Transformer | Difference | Change (%) |
|---|---|---|---|---|
| Alien | **0.142** | 0.061 | 0.081 | **+132.8** |
| Amidar | 0.061 | **0.090** | -0.029 | -32.2 |
| Assault | **1.085** | 0.764 | 0.321 | **+42.0** |
| Asterix | 0.131 | **0.136** | -0.005 | -3.7 |
| BankHeist | 0.557 | **0.998** | -0.441 | -44.2 |
| BattleZone | **0.153** | 0.084 | 0.069 | **+82.1** |
| Boxing | **6.664** | 5.767 | 0.897 | **+15.6** |
| Breakout | **0.881** | 0.458 | 0.423 | **+92.4** |
| ChopperCommand | 0.096 | **0.097** | -0.001 | -1.0 |
| CrazyClimber | **2.400** | 2.041 | 0.359 | **+17.6** |
| DemonAttack | **0.034** | -0.008 | 0.042 | **+525.0** |
| Freeway | **0.582** | 0.174 | 0.408 | **+234.5** |
| Frostbite | 0.044 | **0.171** | -0.127 | -74.3 |
| Gopher | 0.315 | **1.188** | -0.873 | -73.5 |
| Hero | 0.176 | **0.285** | -0.109 | -38.2 |
| Jamesbond | **1.689** | 1.439 | 0.250 | **+17.4** |
| Kangaroo | **1.018** | 0.544 | 0.474 | **+87.1** |
| Krull | **7.019** | 6.846 | 0.173 | **+2.5** |
| KungFuMaster | 1.113 | **1.189** | -0.076 | -6.4 |
| MsPacman | 0.239 | **0.384** | -0.145 | -37.8 |
| Pong | 1.122 | **1.139** | -0.017 | -1.5 |
| PrivateEye | **0.026** | -0.003 | 0.029 | **+966.7** |
| Qbert | 0.146 | **0.344** | -0.198 | -57.6 |
| RoadRunner | 1.251 | **1.731** | -0.480 | -27.7 |
| Seaquest | 0.009 | **0.011** | -0.002 | -18.2 |
| UpNDown | 0.286 | **0.384** | -0.098 | -25.5 |
| **Overall Mean** | **1.048** | 1.012 | **0.036** | **+3.5** |
| **Overall Median** | 0.300 | **0.384** | -0.084 | -21.7 |

Table 2: Performance comparison between STEP-VQ and CB with Transformer on Atari 100K benchmark. Values show mean normalised scores across 3 seeds and evaluate the performance by averaging across 10 episodes. Best performing method per game is highlighted in bold. STEP-VQ has 9 superhuman games (HNS > 1) while CB has 8. Both methods have near-superhuman performances: STEP-VQ's `Breakout` at 0.881 and CB's `BankHeist` at 0.998, achieved with the same hyperparameters across all games.

| Game | STEP-VQ Transformer | IRIS* | Difference | Change (%) |
|---|---|---|---|---|
| Alien | **0.142** | 0.028 | 0.114 | **+407.1** |
| Amidar | 0.061 | **0.080** | -0.019 | -23.8 |
| Assault | 1.085 | **2.504** | -1.419 | -56.7 |
| Asterix | **0.131** | 0.078 | 0.053 | **+67.9** |
| BankHeist | **0.557** | 0.053 | 0.504 | **+950.9** |
| BattleZone | 0.153 | **0.308** | -0.155 | -50.3 |
| Boxing | **6.664** | 5.833 | 0.831 | **+14.2** |
| Breakout | 0.881 | **2.929** | -2.048 | -69.9 |
| ChopperCommand | 0.096 | **0.115** | -0.019 | -16.5 |
| CrazyClimber | **2.400** | 1.938 | 0.462 | **+23.8** |
| DemonAttack | 0.034 | **1.035** | -1.001 | -96.7 |
| Freeway | 0.582 | **1.033** | -0.451 | -43.7 |
| Frostbite | 0.044 | **0.045** | -0.001 | -2.2 |
| Gopher | 0.315 | **0.918** | -0.603 | -65.7 |
| Hero | 0.176 | **0.202** | -0.026 | -12.9 |
| Jamesbond | **1.689** | 1.584 | 0.105 | **+6.6** |
| Kangaroo | **1.018** | 0.263 | 0.755 | **+287.1** |
| Krull | **7.019** | 4.699 | 2.320 | **+49.4** |
| KungFuMaster | **1.113** | 0.957 | 0.156 | **+16.3** |
| MsPacman | **0.239** | 0.104 | 0.135 | **+129.8** |
| Pong | **1.122** | 1.000 | 0.122 | **+12.2** |
| PrivateEye | **0.026** | 0.001 | 0.025 | **+2500.0** |
| Qbert | **0.146** | 0.044 | 0.102 | **+231.8** |
| RoadRunner | **1.251** | 1.226 | 0.025 | **+2.0** |
| Seaquest | 0.009 | **0.014** | -0.005 | -35.7 |
| UpNDown | **0.286** | 0.270 | 0.016 | **+5.9** |
| **Overall Mean** | 1.048 | **1.048** | -0.001 | -0.1 |
| **Overall Median** | **0.300** | 0.289 | **0.011** | **+4.0** |

Table 3: Performance comparison between STEP-VQ Transformer and IRIS on Atari 100K benchmark. Best performing method per game is highlighted in bold. Note: IRIS* results are from Micheli et al. (2023) with different architecture and hyperparameters.

| Game | STEP-VQ Mamba2 | Cross Entropy | Difference | Change (%) |
|---|---|---|---|---|
| Assault | 1.054 | **1.281** | -0.227 | -17.7 |
| BankHeist | **0.294** | 0.287 | 0.007 | **+2.4** |
| Boxing | **7.244** | 5.700 | 1.544 | **+27.1** |
| Breakout | 1.071 | **1.138** | -0.067 | -5.9 |
| Jamesbond | **1.597** | 1.044 | 0.553 | **+53.0** |
| Krull | **7.628** | 5.262 | 2.366 | **+45.0** |
| RoadRunner | **1.528** | 1.469 | 0.059 | **+4.0** |
| UpNDown | 0.203 | **0.376** | -0.173 | -46.0 |
| **Overall Mean** | **2.577** | 2.070 | **0.508** | **+24.5** |
| **Overall Median** | **1.300** | 1.209 | **0.090** | **+7.4** |

Table 4: Performance comparison between STEP-VQ Mamba2 (KL divergence) and Cross Entropy on Atari 100K benchmark. Values show mean normalised scores across 3 seeds. Best performing method per game is highlighted in bold. Percentage change calculated as (STEP-VQ Mamba2 - Cross Entropy) / Cross Entropy $\times$ 100%.

| Game | STEP-VQ HighRes | Categorical HighRes | Difference | Change (%) |
|------|-----------------|---------------------|------------|------------|
| Assault | **1.185** | 0.825 | 0.360 | **+43.6** |
| BankHeist | 0.449 | **0.706** | -0.257 | -36.4 |
| Boxing | **7.075** | 5.156 | 1.919 | **+37.2** |
| Breakout | **0.940** | 0.411 | 0.529 | **+128.7** |
| Jamesbond | **1.689** | 1.220 | 0.469 | **+38.4** |
| Krull | **5.870** | 4.714 | 1.156 | **+24.5** |
| RoadRunner | **1.595** | 1.531 | 0.064 | **+4.2** |
| UpNDown | 0.278 | **0.409** | -0.131 | -32.0 |
| **Overall Mean** | **2.385** | 1.871 | **0.514** | **+27.4** |
| **Overall Median** | **1.390** | 1.022 | **0.368** | **+35.9** |

Table 5: Performance comparison between STEP-VQ Highres and Categorical Highres on Atari 100K benchmark. Values show mean normalized scores across 3 seeds. Best performing method per game is highlighted in bold. Percentage change calculated as (STEP-VQ Highres - Categorical Highres) / Categorical Highres × 100%.

| Game | STEP-VQ HighRes | StepVQ Mamba2 | Difference | Change (%) |
|------|-----------------|---------------|------------|------------|
| Assault | **1.185** | 1.054 | 0.131 | **+12.4** |
| BankHeist | **0.449** | 0.294 | 0.155 | **+52.7** |
| Boxing | 7.075 | **7.244** | -0.169 | -2.3 |
| Breakout | 0.940 | **1.071** | -0.131 | -12.2 |
| Jamesbond | **1.689** | 1.597 | 0.092 | **+5.8** |
| Krull | 5.870 | **7.628** | -1.758 | -23.0 |
| RoadRunner | **1.595** | 1.528 | 0.067 | **+4.4** |
| UpNDown | **0.278** | 0.203 | 0.075 | **+36.9** |
| **Overall Mean** | 2.385 | **2.577** | -0.192 | -7.5 |
| **Overall Median** | **1.390** | 1.300 | **0.091** | **+7.0** |

Table 6: Performance comparison between STEP-VQ Highres and STEP-VQ Mamba2 on Atari 100K benchmark. Values show mean normalized scores across 3 seeds. Best performing method per game is highlighted in bold. Percentage change calculated as (STEP-VQ Highres - STEP-VQ Mamba2) / STEP-VQ Mamba2 × 100%.

| Game | Categorical HighRes | Categorical Mamba2 | Difference | Change (%) |
|------|---------------------|--------------------|------------|------------|
| Assault | **0.825** | 0.737 | 0.088 | **+11.9** |
| BankHeist | 0.706 | **1.371** | -0.665 | -48.5 |
| Boxing | 5.156 | **6.169** | -1.013 | -16.4 |
| Breakout | **0.411** | 0.199 | 0.212 | **+106.5** |
| Jamesbond | **1.220** | 1.190 | 0.030 | **+2.5** |
| Krull | 4.714 | **6.045** | -1.331 | -22.0 |
| RoadRunner | 1.531 | **1.945** | -0.414 | -21.3 |
| UpNDown | 0.409 | **0.963** | -0.554 | -57.5 |
| **Overall Mean** | 1.871 | **2.327** | -0.456 | -19.6 |
| **Overall Median** | 1.022 | **1.280** | -0.258 | -20.1 |

Table 7: Performance comparison between Categorical Highres and Categorical Mamba2 on Atari 100K benchmark. Values show mean normalized scores across 3 seeds. Best performing method per game is highlighted in bold. Percentage change calculated as (Categorical Highres - Categorical Mamba2) / Categorical Mamba2 × 100%.

## F  LOSS AND HYPERPARAMETERS

### F.1  VARIATIONAL AUTOENCODER FOR STEP-VQ-M AND STEP-VQ-T

The hyperparameters shown in Table 8 correspond to the standard model. For both STEP-VQ-M and STEP-VQ-T, we use the same hyperparameter (except for the hyperparameter differences between Mamba-2 and Transformer).

| Hyperparameter | Value |
|---|---|
| Learning rate | 1e-4 |
| Frame shape (h, w, c) | (64, 64, 3) |
| Layers | 4 |
| Filters per layer (Encoder) | (32, 64, 128, 256) |
| Stride | (1, 2, 2, 2) |
| Kernel | 5 |
| Act | SiLU |
| Norm | Group |

Table 8: Hyperparameters for the autoencoder for STEP-VQ-M and STEP-VQ-T.

For the higher resolution experiments, we use the same hyperparameters but with a different frame shape (h, w, c) = (96, 96, 3).

### F.2  VARIATIONAL AUTOENCODER FOR CB-M AND CB-T

The hyperparameters shown in Table 9 correspond to the standard model VAE for CB-M and CB-T, it is also used for higher resolution experiments.

| Hyperparameter | Value |
|---|---|
| Learning rate | 1e-4 |
| Frame shape (h, w, c) | (64, 64, 3) |
| Layers | 5 |
| Filters per layer (Encoder) | (32, 64, 128, 256, 256) |
| Stride | (1, 2, 2, 2, 2) |
| Kernel | 5 |
| Act | SiLU |
| Norm | Group |

Table 9: Hyperparameters for the autoencoder for CB-M and CB-T.

For the higher resolution experiments, we use the same hyperparameters but with a different frame shape (h, w, c) = (96, 96, 3).

### F.3  VECTOR QUANTISATION FOR STEP-VQ-M AND STEP-VQ-T

The vector quantisation for STEP-VQ-M and STEP-VQ-T is the same as shown in Table 10.

| Hyperparameter | Value |
|---|---|
| Codebook size | 64 |
| Commitment loss weight | 0.25 |
| Codebook EMA | 0.99 |

Table 10: Hyperparameters for the vector quantisation for STEP-VQ-M and STEP-VQ-T.

### F.4 CATEGORICAL BOTTLENECK FOR CB-M AND CB-T

The categorical bottleneck for CB-M and CB-T is the same as shown in Table 11.

| Hyperparameter | Value |
| --- | --- |
| Categorical size | 32 |
| Categorical classes | 32 |

Table 11: Hyperparameters for the categorical bottleneck for CB-M and CB-T.

### F.5 REWARD AND TERMINATION PREDICTION HEADS

The reward and termination flag predictors both utilize the deterministic state representation produced by the sequence model. The rich temporal features captured by the sequence model enable accurate predictions using only a single fully connected layer. This hyperparameter is used for all four models: STEP-VQ-M, STEP-VQ-T, CB-M and CB-T.

| Hyperparameter | Value |
| --- | --- |
| Hidden units | 256 |
| Layers | 1 |
| Act | SiLU |
| Norm | RMS |

Table 12: Hyperparameters for reward and termination prediction heads.

### F.6 MAMBA-2 HYPERPARAMETERS

The Mamba-2 hyperparameters are the same as the ones used for STEP-VQ-M and CB-M.

| Hyperparameter | Value |
| --- | --- |
| Learning rate | 4e-5 |
| Hidden units | 512 |
| Layers | 4 |
| Dropout | 0.1 |
| Act | SiLU |
| Norm | RMS |
| Mamba-2 head number | 4 |

Table 13: Hyperparameters for Mamba-2.

### F.7 TRANSFORMER HYPERPARAMETERS

The Transformer hyperparameters are the same as the ones used for STEP-VQ-T and CB-T.

### F.8 ACTOR CRITIC HYPERPARAMETERS

We use the behavior policy learning approach from DreamerV3 (Hafner et al., 2023) due to its simplicity and proven effectiveness, as the behavior policy is not the focus of our main focus in this paper.

| Hyperparameter | Value |
|---|---|
| Learning rate | 4e-5 |
| Hidden units | 512 |
| Layers | 4 |
| Feed forward units | 2048 |
| Dropout | 0.1 |
| Act | SiLU |
| Norm | RMS |
| Transformer head number | 8 |

Table 14: Hyperparameters for Transformer.

| Hyperparameter | Value |
|---|---|
| Learning rate | 4e-5 |
| Layers | 2 |
| Gamma | 0.985 |
| Lambda | 0.95 |
| Entropy coefficient | 3e-4 |
| Max gradient norm | 100 |
| Actor hidden units | 256 |
| Critic hidden units | 512 |
| RMS Norm | True |
| Act | SiLU |
| Batch size ($b_{img}$) | 1024 |
| Imagine context length ($l_{img}$) | 4 |

Table 15: Hyperparameters for the behaviour policy.

# G    RECONSTRUCTED GAMING FRAMES

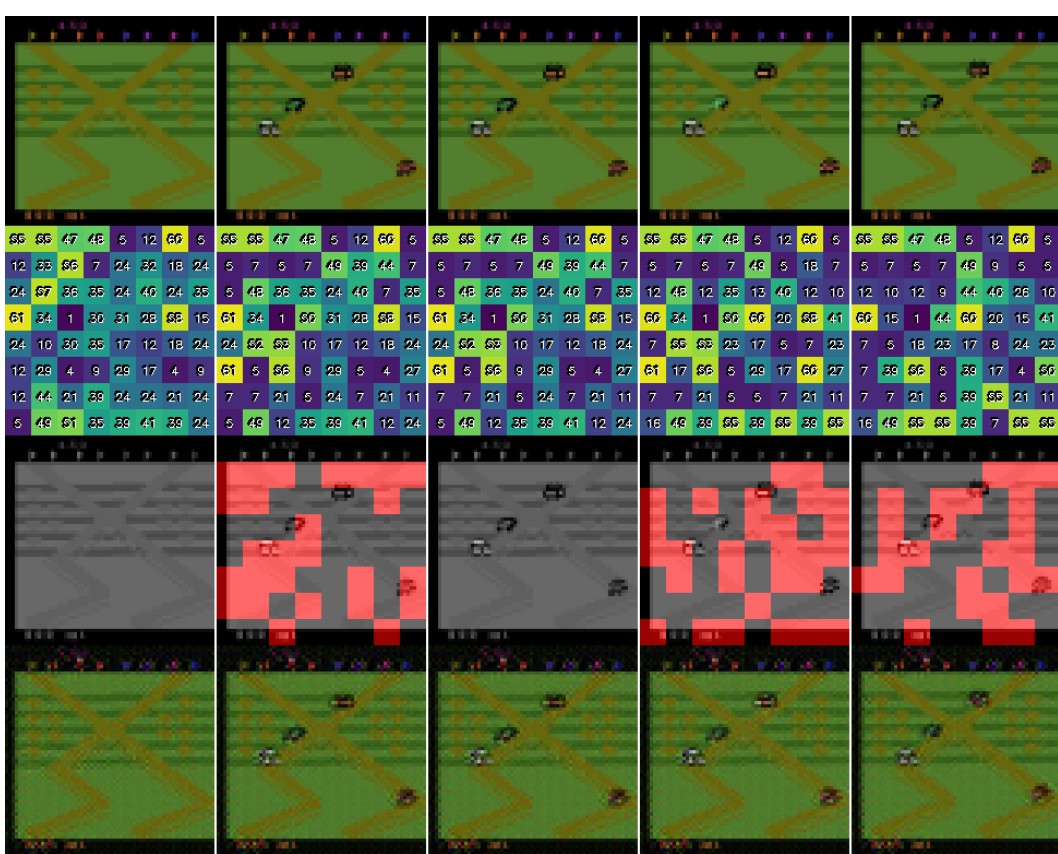

Figure 7: Reconstructed gaming frames from the `UpNDown` game at 64x64 resolution in temporal order. The top row shows the original frames, and the second row shows the quantitised codes. The third row shows the original frames in greyscale at the background and the red patches show where the quantitised codes change from the previous frame. The fourth row is the reconstructed frames.

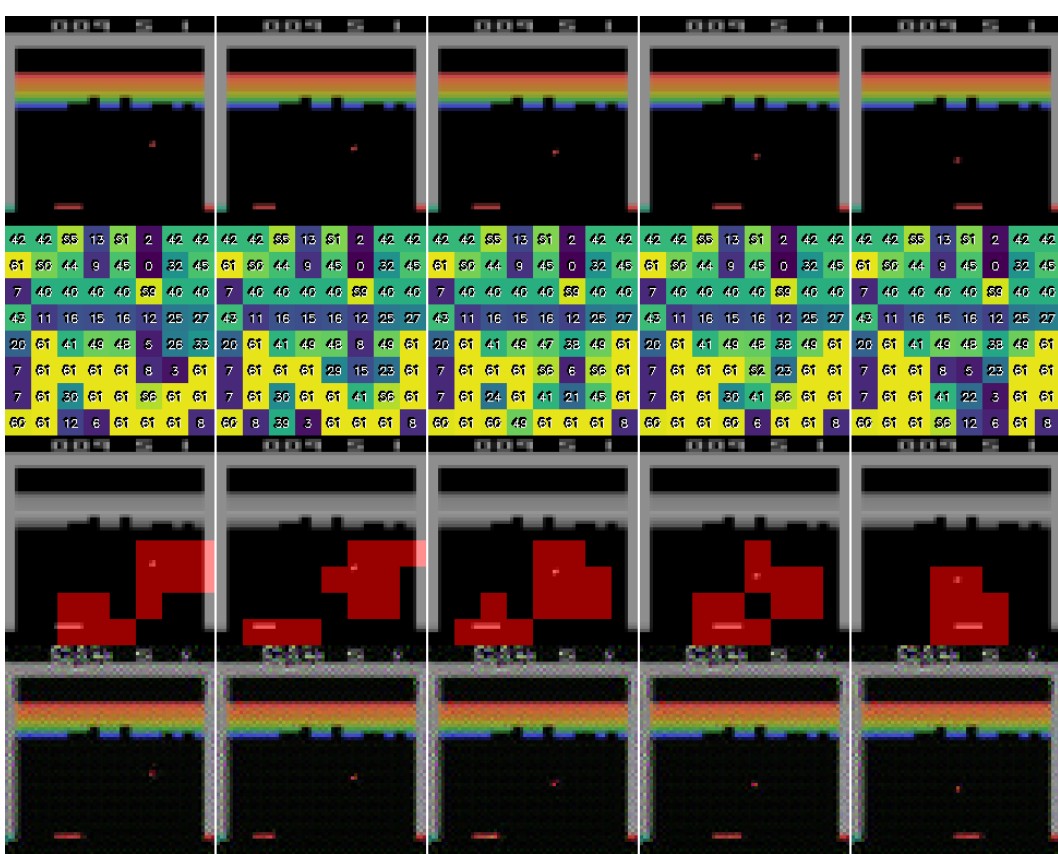

Figure 8: Reconstructed gaming frames from the `Breakout` game at 64x64 resolution in temporal order. The top row shows the original frames, the second row shows the quantitised codes. The third row shows the original frames in greyscale at the background and the red patches show where the quantitised codes change from the previous frame. The fourth row is the reconstructed frames.

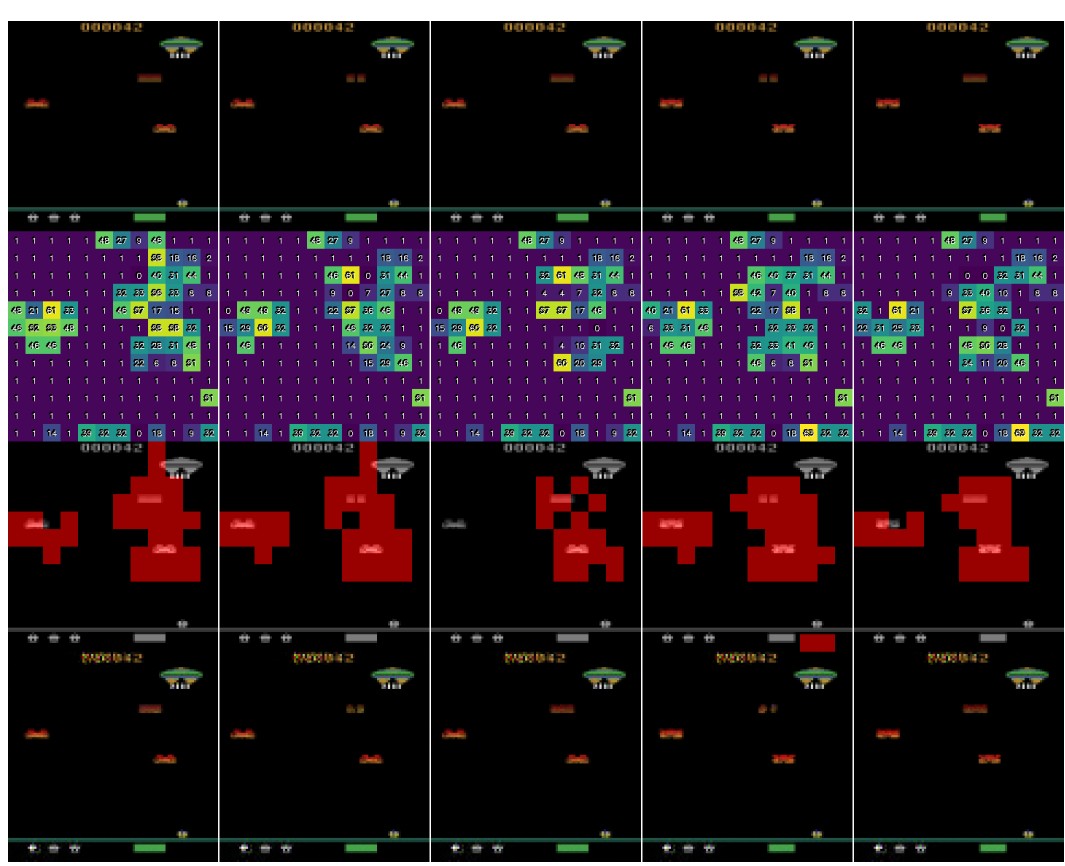

Figure 9: Reconstructed gaming frames from the `Assault` game at 96x96 resolution in temporal order. The top row shows the original frames, the second row shows the quantitised codes. The third row shows the original frames in greyscale at the background and the red patches show where the quantitised codes change from the previous frame. The fourth row is the reconstructed frames.

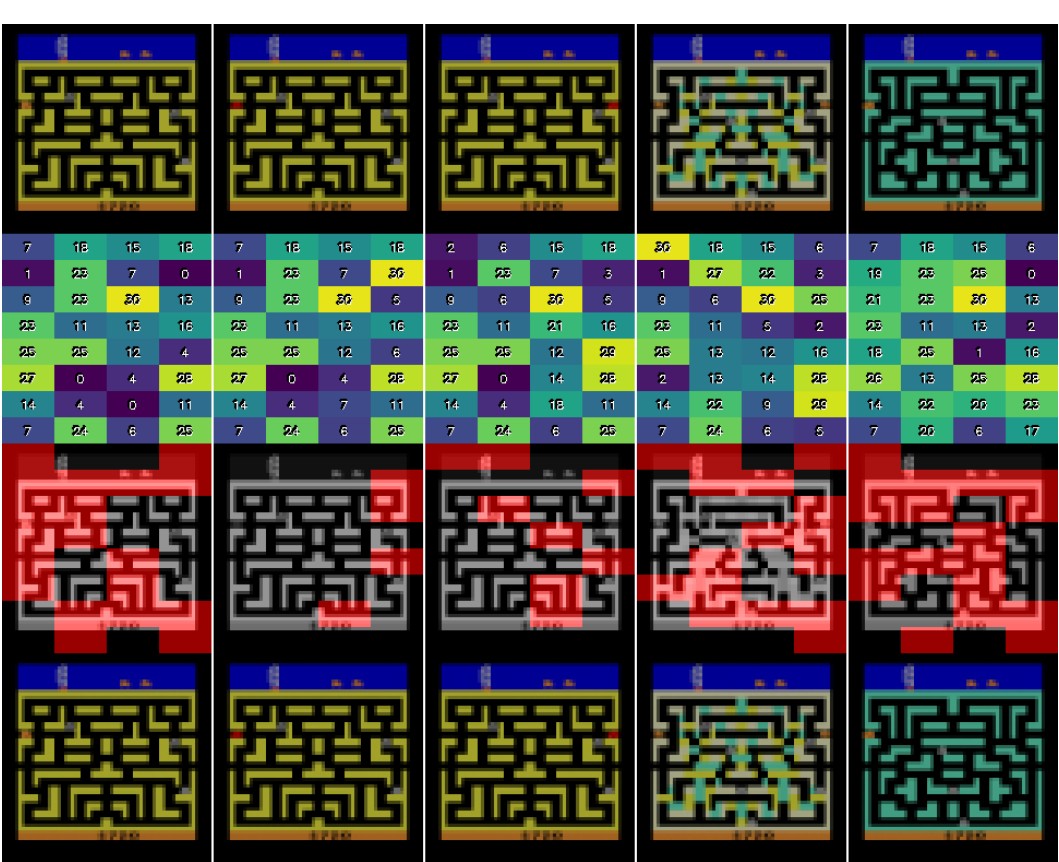

Figure 10: Reconstructed gaming frames from the `BankHeist` game at 64x64 resolution in temporal order. The top row shows the original frames, the second row shows the quantised codes. The third row shows the original frames in greyscale at the background and the red patches show where the quantised codes change from the previous frame. The fourth row is the reconstructed frames.

