# OpenReview forum: "STEP-VQ: Sequence-model Agnostic Frame-level Inference with VQ-VAE for Model-based Reinforcement Learning"
_ICLR.cc/2026/Conference — ICLR 2026 Conference Withdrawn Submission_

### Official Review · Reviewer_zxCC · 2025-10-30

**Soundness:** 2
**Presentation:** 1
**Contribution:** 1
**Rating:** 2
**Confidence:** 4

**Summary:**

The paper introduces a method for efficient world modeling that combines a VQ-VAE autoencoder with a sequence model to enable frame-level prediction of discrete latent representations, avoiding token-level autoregression. The approach models temporal dynamics by predicting the full grid of VQ code distributions for the next frame in a single forward pass and aligns these predictions with encoder posteriors through a KL-divergence-based dynamics loss derived from variational principles. Evaluated on the Atari 100K benchmark, STEP-VQ achieves performance comparable to categorical and autoregressive VQ-based baselines while providing 11× faster training, 4× fewer parameters, and improved scalability at higher resolutions, maintaining efficiency across both Transformer and Mamba-2 sequence model architectures.

**Strengths:**

1. The paper demonstrates a substantial training speedup compared to IRIS-like autoregressive VQ-VAE world models, effectively addressing a known computational bottleneck in VQ-based MBRL.

2. The authors provide a well-structured categorization of existing approaches (categorical bottlenecks vs. VQ-VAE) and clearly identify the trade-offs motivating the proposed method.

3. The inclusion of both Transformer and Mamba-2 sequence models in experiments enhances the robustness of the evaluation and shows the method’s compatibility with different sequence modeling paradigms.

**Weaknesses:**

1. **Unclear methodological exposition:** The description of the proposed approach is mathematically dense and lacks intuitive implementation detail. The role of distributions and losses is not directly linked to concrete architecture design, and the core pipeline (as shown in Figure 6) should appear in the main text to aid understanding.

2. **Limited performance gain over categorical bottlenecks:** While efficiency improves compared with IRIS-like methods, the method does not show clear performance advantages compared to strong CB-based baselines such as DreamerV3 or STORM. Additionally, computational cost comparisons with these CB models are missing, leaving STEP-VQ's relative benefit uncertain.

3. **Insufficient detail in high-resolution comparisons:** The implementation setup of categorical baselines in higher-resolution experiments is under-specified, which weakens the claim that STEP-VQ consistently outperforms CB methods at larger resolutions.

4. **Restricted scalability analysis:** Although the paper highlights resolution scalability, the experiments stop at 96x96 inputs, far below the resolutions addressed by recent models like Dreamer4 [1] (up to 640x360), limiting the evidence for true scalability across visual domains.

[1] Hafner, Danijar, Wilson Yan, and Timothy Lillicrap. "Training agents inside of scalable world models." arXiv preprint arXiv:2509.24527 (2025).

**Questions:**

See the weaknesses

---

> ### Author Response · Authors · 2025-12-04
>
> > The description of the proposed approach is mathematically dense and lacks intuitive implementation detail. The role of distributions and losses is not directly linked to concrete architecture design, and the core pipeline (as shown in Figure 6) should appear in the main text to aid understanding.
>
> We thank the reviewer for this constructive feedback. Whilst the core pipeline is described across Sections 2.1–2.3, we acknowledge that Section 2.3 is mathematically dense and does not explicitly tie each loss term to architectural blocks in a diagrammatic way. In the revised manuscript, we will:
>
> - Move Figure 6 to the main text to provide visual grounding alongside the mathematical formulation.
> - Add a concrete walk-through paragraph that maps each distribution ($p(z_t \mid h_t)$, $q(z_t \mid x_t)$) and loss term to specific network components and data flow during one training step.
>
> > Whilst efficiency improves compared with IRIS-like methods, the method does not show clear performance advantages compared to strong CB-based baselines such as DreamerV3 or STORM. Additionally, computational cost comparisons with these CB models are missing, leaving STEP-VQ's relative benefit uncertain.
>
> We acknowledge that our current submission does not include direct performance comparisons with DreamerV3 or STORM. As noted in our responses to Reviewer 3CbD W1 and VmE4 W1, STEP-VQ is a complementary architectural contribution at the discretisation layer rather than a full competing agent.
>
> >The implementation setup of categorical baselines in higher-resolution experiments is under-specified, which weakens the claim that STEP-VQ consistently outperforms CB methods at larger resolutions
>
> The implementation details for the categorical baseline at higher resolutions are provided in the appendix:
>
> - Appendix F.2 specifies that CB-M and CB-T use the same VAE hyperparameters as at $64 \times 64$, with only the frame shape changed to $(96, 96, 3)$.
> - Appendix F.4 provides the categorical bottleneck hyperparameters.
> - Appendix F.6 provides the Mamba hyperparameters (shared between STEP-VQ-M and CB-M).
>
> To make this more explicit, we will add a clarifying sentence in the main text: “For $96 \times 96$ experiments, we reuse exactly the same CB architecture and hyperparameters, only changing the input resolution.”
>
> > Although the paper highlights resolution scalability, the experiments stop at $96 \times 96$ inputs, far below the resolutions addressed by recent models like Dreamer4 (up to $640 \times 360$), limiting the evidence for true scalability across visual domains.
>
> We thank the reviewer for pointing out recent work such as Dreamer4, which operates at resolutions up to $640 \times 360$. However, we believe this is not a directly comparable setting to our work. Dreamer4 uses a $\sim$2B-parameter diffusion/flow-based world model trained on large offline video datasets, targeting high-capacity continuous world models in domains like Minecraft. In contrast, our focus is on compact, discrete VQ-based world models in the low-data, online Atari-100k regime. The two settings differ in architecture class, parameter scale, and data regime, so we do not claim to match Dreamer4's absolute resolution.
>
> Our “resolution scalability’’ claim is explicitly comparative: we study how STEP-VQ scales relative to categorical bottlenecks (CB) under fixed model families and similar spatial fidelity. CB-style discrete latents are the dominant design in recent state-of-the-art MBRL agents (e.g., Dreamer-family, DRAMA, STORM), where the categorical state is typically flattened and fed into dense recurrent or transformer layers. In these widely used architectures, increasing input resolution whilst preserving latent granularity forces the latent grid to grow, which in turn causes the transition model’s parameter count to scale poorly with resolution. STEP-VQ decouples the codebook size from the latent grid dimensions, allowing us to increase spatial resolution without incurring the same parameter growth in the dynamics model.
>
> Our $64 \rightarrow 96$ experiments are therefore not intended as evidence that STEP-VQ already handles real-world resolutions like $640 \times 360$, but as empirical support that within the Dreamer-style CB regime, STEP-VQ has strictly better resolution scalability: when both methods are scaled up under comparable fidelity, STEP-VQ’s relative advantage over CB widens (+27.4% mean HNS at $96 \times 96$). We will clarify this in the revision by (i) explicitly framing our scalability claim as CB vs STEP-VQ within Atari resolutions, and (ii) softening any language that could be read as claiming demonstrated scalability to very high-resolution visual domains as in Dreamer4.

---

### Official Review · Reviewer_VmE4 · 2025-10-30

**Soundness:** 1
**Presentation:** 2
**Contribution:** 1
**Rating:** 2
**Confidence:** 5

**Summary:**

This paper proposes STEP-VQ, a method combining VQ-VAE with frame-level prediction for model-based RL. While STEP-VQ achieves 11.2× speedup over IRIS and maintains comparable performance on the Atari-100k benchmark with fewer parameters. However, given the mediocre performance, limited baseline comparisons, narrow experimental scope, unvalidated theoretical assumptions, and limited novelty, **I recommend rejection**.

**Strengths:**

The paper proposes replacing cross-entropy with KL divergence for VQ-based methods and demonstrates a 24.5% performance improvement with theoretical justification.

**Weaknesses:**

* The experimental performance is underwhelming. There already exist many model-based RL methods that achieve both faster overall training time and superior performance.
* The paper compares against very limited baselines, I suggest comparing with state-of-the-art VQ-based methods such as Simulus [1], REM [2], and $\Delta$-IRIS [3], as well as other categorical-based methods like TWISTER [4], DyMoDreamer [5], and STORM [6].
* The evaluation is restricted to Atari-100k only. The paper lacks validation on other standard MBRL benchmarks such as Crafter, DeepMind Control Vision.
* The core theoretical assumption $I(z_t[i, j]; z_t[i', j'] | h_t) ≈ 0$ is never empirically measured or validated.
* **Limited Novelty**: Independent prediction of each token in discrete latent world models has already been explored by many works (e.g., REM, Simulus, Transformer World Model [7]), and most categorical-based methods also independently predict each token.

**References**

[1] Uncovering Untapped Potential in Sample-Efficient World Model Agents.

[2] Improving Token-Based World Models with Parallel Observation Prediction.

[3] Efficient World Models with Context-Aware Tokenization.

[4] Learning Transformer-based World Models with Contrastive Predictive Coding

[5] DyMoDreamer: World Modeling with Dynamic Modulation

[6] STORM: Efficient Stochastic Transformer based World Models for Reinforcement Learning

[7] Improving Transformer World Models for Data-Efficient RL

**Questions:**

* **Absolute training time**: What is the actual wall-clock training time (in hours) for a single Atari-100k environment under the experimental settings presented in this paper?
* **Unfair capacity comparison**: STEP-VQ uses a codebook size of 64 while the CB baseline uses categorical classes of 32. Could the performance improvement simply be attributed to this larger representation capacity rather than the method itself?
* **High-resolution scalability insights**: The improved performance at 96×96 resolution seems intuitive given the parameter scaling properties. What unique insights or theoretical understanding do the authors provide beyond the obvious architectural consequence?
* **Vector-based observations**: Would STEP-VQ outperform CB in environments with vector-based observations (e.g., DeepMind Proprio Control) where spatial quantization may not be necessary?
* **Comparison with SOTA**: What advantages does STEP-VQ offer compared to the current state-of-the-art baselines on the Atari-100k benchmark (e.g., TWISTER, EfficientZero V2 [1], EDELINE [2])?

**References**

[1] EfficientZero V2: Mastering Discrete and Continuous Control with Limited Data

[2] EDELINE: Enhancing Memory in Diffusion-based World Models via Linear-Time Sequence Modeling

---

> ### Author Response · Authors · 2025-12-03
>
> > The experimental performance is underwhelming. There already exist many model-based RL methods that achieve both faster overall training time and superior performance.
>
> We thank the reviewer for raising the concern that our “experimental performance is underwhelming’’ compared to recent model-based RL agents such as REM, Simulus, TWISTER, etc.
>
> Our aim in this work is not to propose a new end-to-end Atari-100k agent that is state-of-the-art in absolute score, but to introduce and study a world-model architectural change that can be plugged into existing Dreamer-style agents. STEP-VQ modifies only the discretisation layer (VQ-VAE) whilst keeping both the sequence model and behaviour learning identical to the DRAMA/DreamerV3 family; in that sense, it is complementary rather than competing with full agents such as TWISTER or EfficientZero V2.
>
> We agree that readers may naturally compare our numbers to newer full agents, and in the revision we will (i) make our scope explicit in the introduction and conclusion (“architectural improvement for VQ-based world models, not a new SOTA agent”), and (ii) add a short paragraph in the related-work / discussion section situating our mean HNS and training time qualitatively relative to REM, $\delta$-IRIS, Simulus and other recent methods, emphasising that STEP-VQ’s frame-level VQ dynamics could in principle be combined with those agents’ behaviour and training schemes. We appreciate the reviewer for prompting us to clarify this positioning.
>
> > The paper compares against very limited baselines. Suggest comparing with state-of-the-art VQ-based methods such as Simulus [1], REM [2], and $\delta$-IRIS [3], as well as other categorical-based methods like TWISTER [4], DyMoDreamer [5], and STORM [6].
>
> We thank the reviewer for this suggestion. Our baseline selection was deliberate: our work can be considered as directly modifying the discretisation layer of DRAMA and STORM to determine whether VQ-VAE can serve as a viable alternative to categorical bottlenecks in these models. This is why we constructed CB-M and CB-T as architecture-matched categorical baselines.
>
> Regarding VQ-based methods such as the IRIS family (including $\delta$-IRIS), these systems employ a fundamentally different MBRL pipeline: they reconstruct imagined frames and then encode at the behaviour policy level, rather than learning discrete latent dynamics directly. Therefore, direct performance comparison is not apple-to-apple—the differences would conflate world-model representation choice with behaviour learning architecture.
>
> We will emphasise this positioning more clearly in the future revision, explicitly discussing how STEP-VQ relates conceptually to these methods whilst clarifying why direct performance comparison would confound representation choice with other architectural decisions.
>
> > The evaluation is restricted to Atari-100k only. The paper lacks validation on other standard MBRL benchmarks such as Crafter, DeepMind Control Vision.
>
> We agree with the reviewer that additional evaluation benchmarks would strengthen the paper. However, our current evaluation already requires substantial computational resources: we conduct $4 \times 26 \times 3 = 312$ full training runs across four configurations (STEP-VQ-M, STEP-VQ-T, CB-M, CB-T) on all 26 Atari-100k games with three seeds each, in addition to the high-resolution experiments and ablations reported.
>
> We note that Atari-100k remains a widely accepted benchmark for evaluating model-based RL methods: recent publications including STORM, DRAMA, TWISTER, TWM, and IRIS all use Atari-100k as their primary evaluation benchmark, which showcases its generalisability and relevance to the community. Our controlled comparison on this benchmark provides valuable insights into the VQ vs CB trade-off whilst maintaining consistency with related work.
>
> That said, we acknowledge this limitation and will consider adding evaluation on other benchmarks such as Crafter or DeepMind Control Vision if computational resources become available for the future revision.
>
> > The core theoretical assumption is never empirically measured or validated.
>
> We agree with the reviewer on this important point. The core assumption— that temporal dynamics make spatial independence acceptable for frame-level prediction—deserves empirical support beyond the theoretical argument in Appendix B.2.
>
> In the future revision, we will measure relevant statistics on games where STEP-VQ performs well versus games where it underperforms relative to categorical bottlenecks. This analysis will provide empirical evidence for when and why the spatial independence assumption holds, adding concrete support to the theoretical motivation.

---

> > ### Author Response · Authors · 2025-12-03
> >
> > > Independent prediction of each token in discrete latent world models has already been explored by many works (e.g., REM, Simulus, Transformer World Model [7]), and most categorical-based methods also independently predict each token.
> >
> > We thank the reviewer for raising this important point about novelty. Whilst we acknowledge conceptual similarities to prior work in parallel prediction, there are fundamental architectural differences that we will clarify more explicitly in the revision.
> >
> > The key distinction lies in the representation structure: REM’s POP, Simulus, and Transformer World Model are Token-Based World Models (TBWMs) that operate on 1D sequences of tokens, where each token represents abstract semantic units. In contrast, STEP-VQ uses VQ-VAE’s 2D spatial grids ($H' \times W'$) of discrete codes that preserve spatial locality through codebook quantisation. Our novelty is enabling frame-level prediction across these 2D spatial grids through spatial independence factorisation, eliminating the need for intra-frame autoregressive prediction whilst maintaining VQ-VAE’s parameter efficiency.
> >
> > Regarding categorical bottlenecks: whilst CB methods like Dreamer do factorise their priors, they flatten spatial features into categorical distributions, completely discarding spatial structure. STEP-VQ maintains the 2D spatial lattice through codebook quantisation, preserving more spatial information than CB whilst achieving comparable efficiency.
> >
> > The paper does mention REM/POP (Section 4.2) but we acknowledge we could be more explicit about these architectural differences. In the revision, we will add a detailed comparison clarifying: (1) TBWM 1D token sequences vs VQ-VAE 2D spatial grids, (2) how our spatial independence assumption differs from token-level independence in TBWMs, (3) why this enables the unique combination of VQ parameter efficiency with frame-level prediction speed, and (4) critically, POP requires modification of the sequence model architecture (using RetNet’s modified forward mode), whereas STEP-VQ is sequence-model agnostic and can be directly applied to existing methods like STORM, DRAMA, or Dreamer by simply replacing the discretisation layer without any sequence model modifications.
> >
> > > What is the actual wall-clock training time (in hours) for a single Atari-100k environment under the experimental settings presented in this paper?
> >
> > We thank the reviewer for this question. We will add specific wall-clock training times for STEP-VQ and the categorical baselines in the future revision.
> >
> > > STEP-VQ uses a codebook size of 64 whilst the CB baseline uses categorical classes of 32. Could the performance improvement simply be attributed to this larger representation capacity rather than the method itself?
> >
> > We thank the reviewer for this insightful question. This capacity comparison actually demonstrates STEP-VQ’s core advantage over categorical bottlenecks rather than representing unfairness.
> >
> > The key distinction lies in parameter scaling: CB parameter counts grow with both categorical class count and the number of categories ($M \times K$), whilst STEP-VQ’s codebook parameters remain fixed regardless of spatial resolution. STEP-VQ uses a codebook size of 64, whereas CB uses $32 \times 32$ (32 categorical positions, each with 32 classes). If we were to increase CB to match STEP-VQ’s representation capacity with $64 \times 64$ categorical positions and classes, the CB parameter count would scale by approximately $4\times$, making it prohibitively expensive.
> >
> > In the current settings, STEP-VQ and CB have comparable overall model sizes. However, if we were to match the codebook sizes exactly—either by increasing CB to $64 \times 64$ or reducing STEP-VQ to $32$—this would create an unfair parameter comparison: the former would drastically increase CB’s already larger parameter count, whilst the latter would reduce STEP-VQ’s representation capacity. The current configuration demonstrates that STEP-VQ achieves competitive performance whilst maintaining parameter efficiency through its architectural design.

---

> > > ### Author Response · Authors · 2025-12-03
> > >
> > > > The improved performance at $96 \times 96$ resolution seems intuitive given the parameter scaling properties. What unique insights or theoretical understanding do the authors provide beyond the obvious architectural consequence?
> > >
> > > We thank the reviewer for raising the question of whether our $96 \times 96$ results provide insight beyond “obvious parameter scaling.” Our claim is not that CB cannot keep parameters constant with higher input resolution in principle, but that in practically effective RL architectures this comes at a substantial cost in representation quality.
> > >
> > > Concretely, in the Dreamer/DRAMA/STORM-style CB baselines we build on, the discrete state is obtained by encoding the image into an $H' \times W'$ grid and then flattened into a vector that is processed by dense recurrent layers. If we want to maintain a fixed spatial granularity (e.g., roughly 1 latent token per $8 \times 8$ pixels), then increasing input resolution necessarily increases the number of latent tokens. In these standard flattened CB architectures, this causes the input dimension—and therefore the parameter count of the transition model—to grow superlinearly with resolution. It is true that one could control parameters by increasing encoder stride and shrinking $H' \times W'$, but this “aggressive downsampling’’ compresses a larger visual field into the same number of bits, discarding exactly the small, high-frequency details (bullets, balls, projectiles) that Atari-style RL depends on.
> > >
> > > STEP-VQ is designed to address this fidelity vs parameter trade-off: the VQ codebook size is independent of $H'$ and $W'$, so we can increase the latent grid resolution to preserve spatial detail without incurring a proportional growth in the number of parameters in the dynamics model. Our $96 \times 96$ experiments empirically support this argument: when we scale both methods in a way that preserves useful spatial information, STEP-VQ's relative advantage over CB grows to $+27.4\%$ mean HNS. We will clarify this in the paper by explicitly stating that our scalability claim is made under the realistic constraint of maintaining comparable spatial fidelity, not under an unconstrained architectural search where CB could be redesigned to be fully convolutional.
> > >
> > > > Would STEP-VQ outperform CB in environments with vector-based observations (e.g., DeepMind Proprio Control) where spatial quantisation may not be necessary?
> > >
> > > No. STEP-VQ is specifically designed for visual observations where spatial structure is meaningful and can be exploited through 2D VQ grids. For vector-based proprioceptive observations where spatial structure is absent, CB or continuous latent representations would be more natural and appropriate. The core advantage of STEP-VQ—preserving spatial locality through 2D quantisation—does not apply to such domains.
> > >
> > > > What advantages does STEP-VQ offer compared to the current state-of-the-art baselines on the Atari-100k benchmark (e.g., TWISTER, EfficientZero V2, EDELINE)?
> > >
> > > As discussed in our response to W1, our aim is not to propose a new state-of-the-art end-to-end agent, but to introduce an architectural improvement to the discretisation layer that can be plugged into existing Dreamer-style agents. STEP-VQ is complementary rather than competing with full agents such as TWISTER, EfficientZero V2, and EDELINE, which represent different research directions (e.g., search-based planning, novel training procedures) orthogonal to our focus on the VQ vs CB tradeoff. In a future revision, we will provide a qualitative comparison situating STEP-VQ relative to these methods whilst making explicit that our contribution targets the frame-level discretisation design.

---

### Official Review · Reviewer_3CbD · 2025-10-31

**Soundness:** 2
**Presentation:** 1
**Contribution:** 3
**Rating:** 2
**Confidence:** 4

**Summary:**

This paper introduces STEP-VQ, a model-based reinforcement learning approach combining vector-quantized VAEs with sequence models to enable frame-level prediction instead of slower token-level autoregression. The key idea is that temporal redundancy between frames can compensate for the loss of fine-grained spatial modelling, thus achieving both efficiency and scalability. The authors claim large speedups over prior VQ-based methods (e.g., IRIS), improved parameter efficiency over categorical bottlenecks, and competitive performance on Atari 100K using both Mamba-2 and Transformer architectures.

**Strengths:**

- This paper replaces token-level autoregression with frame-level prediction, yielding large training speedups (up to ~11x) while keeping VQ-VAE parameter efficiency.
- The method works with both Mamba-2 and Transformers, showing comparable performance without architecture-specific changes.

**Weaknesses:**

- The dynamics model remains unclear. In Section 2.3.1, the same function $f_\psi$ appears to serve multiple roles: in Eq. (3), it computes a hidden vector from latent codes $z_t$, but in Eq. (4), it seems to output a 3D tensor of logits indexed spatially. Even Figure 6(b) does not fully clarify the mapping between these representations. Two possible interpretations are: (i) $f_\psi$ maps between vector and 3D tensor representations through convolutional and transposed-convolutional layers (less likely), or (ii) each latent code is processed independently by the same model (more likely, but the notation in Section 2.3.1 would then be inconsistent). This ambiguity should be clarified.
- Line 93: The statement about "explosive parameter scaling" may be overstated. The scaling depends on whether encoder output dimensions $H', W'$ grow with input size $H, W$, since additional downsampling could mitigate the effect. Clarifying this assumption would improve accuracy.
- The KL balancing loss resembles the one in DreamerV2 (Hafner et al., 2020, Algorithm 2). This work should be cited explicitly, and the differences between the two formulations should be discussed. Also, the equation in line 253 lacks a label.
- Evaluation setup:
  * The abstract states: "STEP-VQ reaches superhuman performance on 9 games versus 8 for categorical methods, with KL divergence providing 24.5% improvement over cross-entropy baselines." While this highlights per-game counts and a within-method loss comparison, Tables 1-2 show only small mean gains alongside notably lower medians (Mamba: mean +5.4%, median -35.6%; Transformer: mean +3.5%, median -21.7%), suggesting heavier-tailed outcomes and wins concentrated in fewer games. I recommend rephrasing the claim to reflect these results, and explicitly clarifying that the "24.5% improvement" refers to KL vs. cross-entropy **within** STEP-VQ, not versus categorical baselines.
  * Only three seeds per game are used, which is insufficient given the known high variance in Atari 100K. At least five seeds are typically considered a minimum.
  * Section 3 does not clearly state which architectures (Mamba or Transformer) are used in the results being compared. This makes it impossible to compare to the other results at lower resolution.
  * Figure 3 would be more informative if it included the lower-resolution baselines, as done in Tables 6-7. This would better illustrate scalability trends across resolutions.
- A relevant related work is missing: Robine et al. "Smaller World Models for Reinforcement Learning." Neural Process Lett 55, 11397–11427 (2023).
- Minor issues:
  * Line 156: The encoder is referred to as $Enc_\phi$ here but later as the probabilistic model $q_\phi$. The relationship between these should be clarified or unified in notation.
  * Line 191: The additional variable $z^\star_t$ seems unnecessary. Reusing $z^q_t$ would simplify the presentation.
  * Line 196: The indexing notation (e.g., [:, 1:L]) looks like Python slicing, which implies exclusive upper bounds. Using $z^\star_{2:L}$ and $\hat{z}_{1:L-1}$, consistent with 1-based time indexing elsewhere, would improve clarity.
  * Figure 2(a): The legend is confusing.

Overall, the paper makes a promising step toward more efficient world models, but the technical description of the dynamics model and evaluation methodology need clarification before publication. Further experiments (e.g., with more seeds or RNN variants) would strengthen the empirical case.

**Questions:**

- Have the authors considered evaluating a recurrent (RNN-based) dynamics model?
- Please clarify the dynamics model as described in the weaknesses.

---

> ### Author Response · Authors · 2025-12-03
>
> >The dynamics model remains unclear. In Section 2.3.1, the same function appears to serve multiple roles: in Eq. (3), it computes a hidden vector from latent codes, but in Eq. (4), it seems to output a 3D tensor of logits indexed spatially. Even Figure 6(b) does not fully clarify the mapping between these representations. Two possible interpretations are: (i) maps between vector and 3D tensor representations through convolutional and transposed-convolutional layers (less likely), or (ii) each latent code is processed independently by the same model (more likely, but the notation in Section 2.3.1 would then be inconsistent).
> We acknowledge this valid concern about notation clarity. The confusion arises because we overloaded the symbol $f_\psi$ to represent both the recurrent dynamics core and the subsequent mapping to spatial logits. We will revise the notation as follows:
>
> We acknowledge this valid concern about notation clarity. The confusion arises because we overloaded the symbol $f_\psi$ to represent both the recurrent dynamics core and the subsequent mapping to spatial logits. We will revise the notation as follows:
>
> - **Recurrent core:** $f_\psi$ maps the recurrent state: $h_t = f_\psi(h_{t-1}, z_{t-1}, a_{t-1})$, where $h_t \in \mathbb{R}^{d_h}$ (Eq. 3).
>
> - **Output head:** We introduce $g_\phi$ to explicitly denote the output head that maps the recurrent state to spatial logits: $g_\phi(h_t) \in \mathbb{R}^{H' \times W' \times C}$, producing a grid of logits over codebook indices for each spatial location (Eq. 4).
>
> The complete pipeline is: the sequence model $f_\psi$ outputs states of shape $(B, L, d_h)$; then the output head $g_\phi$ independently processes each $h_t$ to produce a $(H', W', C)$ tensor of categorical distributions over codes. Each spatial position $(i,j)$ is predicted independently given $h_t$.
>
> We will update Section 2.3.1 with this notation and enhance Figure 6(b) to explicitly show both components and their shapes.
>
> > The statement about ``explosive parameter scaling'' may be overstated. The scaling depends on whether encoder output dimensions grow with input size, since additional downsampling could mitigate the effect. Clarifying this assumption would improve accuracy.
>
> Our 'explosive scaling' statement is made under a fixed-fidelity assumption: the latent grid size grows with input resolution instead of collapsing to a tiny vector. Our CB baselines are constructed in this regime to preserve more spatial structure than the heavily compressed latents used in Dreamer-style agents. Under this design, flattening the grid into a single vector for the recurrent dynamics leads to parameter counts that grow linearly with the number of tokens (i.e., with image area).
>
> >The KL balancing loss resembles the one in DreamerV2 (Hafner et al., 2020, Algorithm 2). This work should be cited explicitly, and the differences between the two formulations should be discussed. Also, the equation in line 253 lacks a label.
>
> We acknowledge this valid point. Our KL balancing approach follows the spirit of DreamerV2's KL balancing trick (Hafner et al., 2020, Algorithm 2), where a symmetric KL divergence with stop-gradients on each side is used. We will add an explicit citation to DreamerV2 and provide an equation label for the KL loss.
>
> The key difference is that we apply this balancing between encoder posteriors and discrete dynamics priors, specifically using the distance between the encoder output and the VQ codebook as the posteriors, whereas DreamerV2 applies it between continuous latent distributions.
>
> >The abstract states: ''STEP-VQ reaches superhuman performance on 9 games versus 8 for categorical methods, with KL divergence providing 24.5\% improvement over cross-entropy baselines.'' ...
>
> We acknowledge this concern. As we need to revise our results to incorporate new improvements, the phrasing here will change. We will ensure that in the future revision, we will be more specific about the comparison context and statistical characteristics of the results.

---

> > ### Author Response · Authors · 2025-12-03
> >
> > >Only three seeds per game are used, which is insufficient given the known high variance in Atari 100K. At least five seeds are typically considered a minimum.
> >
> > On the number of seeds. We agree that Atari 100k exhibits high variance and that, in an ideal setting, more than three seeds per game would be preferable. However, our current evaluation already requires substantial compute: we run four configurations across all 26 Atari 100k games with three seeds each ($4 \times 26 \times 3 = 312$ full training runs), in addition to the high-resolution experiments and ablations reported in the paper. Scaling all configurations to five seeds would increase the cost significantly and is not feasible under our current budget.
> >
> > Instead, in the revised version we follow the evaluation methodology of Agarwal et al., *Deep RL at the Edge of the Statistical Precipice* (NeurIPS 2021), which is explicitly designed for the few-run regime (3--10 runs). Concretely, we now report interquartile mean (IQM) scores, stratified bootstrap 95% confidence intervals, and performance distributions across games, rather than relying solely on point estimates. This provides a more statistically robust comparison whilst staying within realistic computational limits.
> >
> > If additional computational resources become available, we will prioritise adding two extra seeds for the main STEP-VQ and categorical baselines to reach five seeds on Atari 100k and will include those results in an updated version.
> >
> > > Section 3 does not clearly state which architectures (Mamba or Transformer) are used in the results being compared. This makes it impossible to compare to the other results at lower resolution.
> >
> > We respectfully disagree with this assessment. The paper explicitly states which architectures are used throughout Section 3:
> >
> > - **Section 3 (opening):** “Unless otherwise stated, results use the configurations in Sec. F for the Mamba2-based (STEP-VQ-M) and Transformer-based (STEP-VQ-T) architectures.”
> >
> > - **Section 3.1:** Explicitly identifies that STEP-VQ-IRIS uses a Transformer, and STEP-VQ-T is “the configuration used for performance evaluation in Table 3.”
> >
> > - **Section 3.2:** Clearly names all method variants: “STEP-VQ-M, STEP-VQ-T, CB-M, CB-T” and states that “all use identical hyperparameters across all Atari games.”
> >
> > - **Section 3.3:** While the higher-resolution experiments do not repeat the backbone architecture explicitly in the main text, Tables 5–7 in the appendix clearly show the method names (STEP-VQ HighRes uses the Mamba2 backbone as indicated by comparison with STEP-VQ Mamba2 in Table 6), and the hyperparameter tables in Appendix F confirm that the same configurations are used with only the input resolution changed.
> >
> > We acknowledge that Section 3.3 could benefit from an additional explicit statement about the architecture used (“using Mamba2-based models”), and we will add this clarification in the revised version. However, the information is present and traceable throughout the paper.
> >
> > > Figure 3 would be more informative if it included the lower-resolution baselines, as done in Tables 6-7. This would better illustrate scalability trends across resolutions.
> >
> > We agree this is a valid suggestion. Currently, Figure 3 shows only the $96 \times 96$ resolution results, whilst Tables 6--7 provide the quantitative comparison between $64 \times 64$ and $96 \times 96$ resolutions. Adding the lower-resolution baselines (64×64 curves or at least baseline reference points) to Figure 3 would indeed make the scalability story more visually obvious and help readers immediately grasp the resolution-dependent performance trends.
> >
> > We will update Figure 3 in the revised version to include the $64 \times 64$ performance curves alongside the $96 \times 96$ results, making the improved scalability of STEP-VQ relative to categorical methods more immediately apparent.
> >
> > >A relevant related work is missing: Robine et al. ``Smaller World Models for Reinforcement Learning.'' Neural Process Lett 55, 11397--11427 (2023).
> >
> > We thank the reviewer for pointing out this relevant work. We will add Robine et al. (2023) to the related work section in the future revision.
> >
> > >Minor Issues
> >
> > We thank the reviewer for pointing out these notation and presentation issues. We will address all of them in the future revision.

---

> > > ### Author Response · Authors · 2025-12-03
> > >
> > > > Have the authors considered evaluating a recurrent (RNN-based) dynamics model?
> > >
> > > Our primary goal was to demonstrate that STEP-VQ is architecture-agnostic across modern sequence models rather than to provide an exhaustive benchmark over all possible backbones. Adding a full LSTM/GRU baseline across all 26 Atari games and configurations would require substantial additional compute beyond the existing $4 \times 26 \times 3$ runs and high-resolution experiments already reported.
> > >
> > > Nothing in STEP-VQ is specific to Transformers or Mamba: the same dynamics loss and frame-level prediction apply directly to standard RNNs. If additional computational resources become available, we would be happy to include an RNN variant (e.g., GRU-based) in a future revision to further illustrate this compatibility.

---

### Author Response · Authors · 2025-12-04
**Withdrawal Notice**

We thank all three reviewers for their thorough and constructive feedback. An update of our work after submission of the paper, we have conducted additional experiments and analysis that have yielded further findings requiring substantial revision of our work. These new results strengthen our claims and address several of the reviewers' concerns regarding performance comparisons with categorical bottlenecks.

Given the significance of these improvements, we have decided to **withdraw this submission** and to submit a revised version that incorporates:

- Improved experimental results demonstrating clearer advantages over categorical bottlenecks on both mean and median scores
- Additional empirical validation of our theoretical assumptions
- Enhanced architectural clarifications and methodological exposition
- Expanded experimental analysis including training time comparisons and capacity studies

We believe these revisions will substantially strengthen the paper and better address the valuable concerns raised by the reviewers. We appreciate the reviewers' time and expertise, and we look forward to submitting an improved version of this work.

---

### Note · Authors · 2025-12-04

**Comment:**

We thank all three reviewers for their thorough and constructive feedback. After the submission of this work, we have some important updates. We have conducted additional experiments and analysis that have yielded further findings requiring substantial revision of our work. These new results strengthen our claims and address several of the reviewers' concerns regarding performance comparisons with categorical bottlenecks.

Given the significance of these improvements, we have decided to **withdraw this submission** and to submit a revised version that incorporates:

- Improved experimental results demonstrating clearer advantages over categorical bottlenecks on both mean and median scores
- Additional empirical validation of our theoretical assumptions
- Enhanced architectural clarifications and methodological exposition
- Expanded experimental analysis including training time comparisons and capacity studies

We believe these revisions will substantially strengthen the paper and better address the valuable concerns raised by the reviewers. We appreciate the reviewers' time and expertise, and we look forward to submitting an improved version of this work.

**Withdrawal Confirmation:**

I have read and agree with the venue's withdrawal policy on behalf of myself and my co-authors.